# Cooperative interaction between ERα and the EMT-inducer ZEB1 reprograms breast cancer cells for bone metastasis

Nastaran Mohammadi Ghahhari [1], Magdalena K. Sznurkowska[2], Nicolas Hulo[3], Lilia Bernasconi[1], Nicola Aceto[2] & Didier Picard [1✉]

The epithelial to mesenchymal transition (EMT) has been proposed to contribute to the metastatic spread of breast cancer cells. EMT-promoting transcription factors determine a continuum of different EMT states. In contrast, estrogen receptor α (ERα) helps to maintain the epithelial phenotype of breast cancer cells and its expression is crucial for effective endocrine therapies. Determining whether and how EMT-associated transcription factors such as ZEB1 modulate ERα signaling during early stages of EMT could promote the discovery of therapeutic approaches to suppress metastasis. Here we show that, shortly after induction of EMT and while cells are still epithelial, ZEB1 modulates ERα-mediated transcription induced by estrogen or cAMP signaling in breast cancer cells. Based on these findings and our ex vivo and xenograft results, we suggest that the functional interaction between ZEB1 and ERα may alter the tissue tropism of metastatic breast cancer cells towards bone.

[1] Département de Biologie Cellulaire, Université de Genève, Sciences III, 1211 Genève 4, Switzerland. [2] Department of Biology, Institute of Molecular Health Sciences, ETH Zurich, 8093 Zürich, Switzerland. [3] Institute of Genetics and Genomics of Geneva, Université de Genève, 1211 Genève 4, Switzerland. ✉email: didier.picard@unige.ch

ERα is a nuclear hormone receptor that mediates the transcriptional regulation of specific target genes during normal mammary development and breast tumorigenesis[1,2]. Because ERα drives two-thirds of breast cancers, it has been recognized as an important prognostic marker and a therapeutic target. Using potent ERα antagonists such as tamoxifen and fulvestrant (ICI), ERα-positive (ERα+) breast tumors are targeted with antiestrogen therapy. However, more than a quarter of all breast cancer patients develop antiestrogen resistance, which remains a major hurdle in managing their clinical outcome[3–5]. Among a plethora of mechanisms that have been found to contribute to endocrine resistance, there are distinct changes in the tumor microenvironment, which stimulate cancer cell proliferation and induce invasiveness[6,7]. In this context, epithelial–mesenchymal transition (EMT) of non-invasive breast cancer cells has been proposed to play a key role in their progression to high-grade metastatic tumors and differential responses to endocrine therapy[5,8–10].

EMT is orchestrated through the action of several transcription factors (EMT–TFs), which shape the malignant transformation of carcinoma cells by modifying gene expression[11–13]. ZEB1/2, SNAIL1/2, and TWIST1/2 are core EMT–TFs, which regulate the transitions among different EMT stages in an interdependent fashion[6,8,14]. A substantial set of genes involved in the maintenance of the epithelial state (for example the E-cadherin gene CDH1) are repressed upon activation of EMT–TFs. In parallel, with the activation of genes associated with the mesenchymal state, a partial mesenchymal phenotype is acquired[15,16]. However, recent studies suggest that a partial and reversible EMT phenotype or an intermediate hybrid state of breast cancer cells is associated with metastasis, chemoresistance, and poor prognosis for the patients[11,17–19].

ZEB1 is a key factor for cell fate determination, tumor initiation, cancer cell plasticity, and metastatic dissemination[14,20,21]. ZEB1 is generally considered to be a transcriptional repressor, but it can also act as a transcriptional activator[22,23]. Compared to luminal breast cancer subtypes, ZEB1 is highly expressed in triple-negative breast cancers, which express neither ERα nor progesterone receptor (PR), which is encoded by an ERα target gene[24,25]. Comprehensive analyses of samples from breast cancer patients support the coexistence of epithelial cells with low levels of ERα with mesenchymal-like cells expressing high levels of ERα in the same tumor microenvironment[18]; the latter are reminiscent of cells in an EMT hybrid state. Although the loss of ERα function promotes an EMT-associated phenotype in breast cancer cells[26–28], ERα activation can also induce EMT in other hormone-inducible cancers[12,29].

The transcriptional activity of ERα can be switched on by both cognate ligand and ligand-independent pathways to regulate cell functions in the mammary epithelium[30]. The binding of 17β-estradiol (E2) to its hormone-binding domain (HBD) triggers the binding of an ERα homodimer to specific DNA sequences containing estrogen response elements (EREs), often dependent on the prior binding of pioneer factors such as FOXA1, GATA3, and AP2γ to chromatin[4,31–33]. cAMP-activated protein kinase A (PKA) activates ERα primarily indirectly by promoting the phosphorylation of ERα coregulators including CARM1, LSD1, CREB1, and their interactions with ERα[34–37].

Here we investigate the effects of the EMT-inducer ZEB1 on both liganded and unliganded ERα transcriptional responses. Indeed, it was unknown whether the two factors cooperate to modulate EMT programs in breast cancer and to transform non-metastatic into more invasive cancer cells. The discovery of such mechanisms may reveal molecular targets that could lead to more effective therapeutic strategies to prevent breast cancer progression. By analyzing the ZEB1-ERα interdependent transcriptional activities, we reveal new mechanisms by which ZEB1 drives tumor progression and invasion of ERα+ breast cancer cells.

## Results

**ZEB1 enhances ERα transcriptional activity during early EMT stages.** We used the luminal breast cancer cell line MCF7 and its variant MCF7-V, which displays more robust ERα responses, and the luminal breast cancer cell line T-47D to establish cells stably expressing ZEB1 from a doxycycline (DOX)-inducible (Tet-on) lentiviral vector. We monitored the expression of EMT-associated markers in the absence (−DOX) or presence (+DOX) of DOX after short-term (1–2 weeks) and long-term (8–12 weeks) expression of ZEB1 to achieve partial and complete EMT, respectively. A complete EMT indicated by the expression of mesenchymal markers vimentin and N-cadherin was detected after long-term expression of ZEB1 (Fig. 1a and Supplementary Fig. 1a). ZEB1 downregulated ERα expression, in agreement with previous reports[38,39]. Surprisingly, 1–2 weeks after induction of ZEB1 expression, ERα expression was still maintained (Fig. 1a). Because ZEB1 did not affect ERα levels in the short-term, we wondered whether it affects ERα transcriptional activity. This was explored with luciferase reporter assays with various cell lines. ZEB1 significantly enhanced an ERE-containing luciferase reporter activity upon activation of ERα by E2 or by increased levels of intracellular cAMP (Fig. 1b, c, and Supplementary Fig. 1b). cAMP levels were induced by treating the cells with forskolin to stimulate adenylate cyclase and with 3-isobutyl-1-methylxanthine to block phosphodiesterase, a cocktail which we will abbreviate as FI. We monitored the short- and long-term effects of ZEB1 on ERα activity in MCF7-V-ZEB1 cells (Supplementary Fig. 1c, d). Short-term expression of ZEB1 enhanced ERα activity (Supplementary Fig. 1c), while the prolonged induction of ZEB1 expression reduced ERα activity (Supplementary Fig. 1d), consistent with the downregulation of ERα levels (Fig. 1a and Supplementary Fig. 1a). Other EMT–TFs, TWIST1, and PRRX1[40], repressed the ERα transcriptional response; however, expression of ZEB1 reversed this effect and increased ERα activity (Fig. 1d). Because ZEB1 expression also correlates with the presence of other hormone receptors[41], we used a construct containing the progesterone response element (PRE-Luc) to measure the PR activity. In contrast to ERα, PR activity was strongly repressed by ZEB1 (Fig. 1e and Supplementary Fig. 1e), indicating a specific ERα-dependent response to ZEB1.

ZEB1 repressed the E-cadherin promoter and stimulated the transcription from the vascular endothelial growth factor A (VEGFA) promoter as expected[42,43], the latter being more strongly activated in the presence of ERα (Fig. 1f, g). Transforming growth factor β (TGFβ) induces the EMT[8], and SMAD4 is a TGFβ-mediating transcriptional co-repressor for ERα in breast cancer[44]. Although TGFβ alone repressed ERα activity, ZEB1 significantly reversed this effect (Fig. 1h). ZEB1 did not affect the activation of a SMAD reporter (Fig. 1i), confirming that the ERE is essential for ZEB1-mediated enhanced ERα activity. To determine if ZEB1 affects the expression of ERα target genes, we examined their mRNA levels. ZEB1 increased expression levels of all assessed genes (Fig. 1j and Supplementary Fig. 1f, g). Conversely, long-term expression of ZEB1, resulting in a mesenchymal-like phenotype, reduced the expression of ERα targets (Supplementary Fig. 1h). Overall, these findings indicate that ZEB1 potentiates ERα-mediated transcription in ligand-dependent and -independent manners, possibly with functional relevance to the early/hybrid stages of EMT.

**Increased ZEB1-induced invasion ability of breast cancer cells is associated with ERα.** Activation of ERα by E2 increases breast cancer cell invasiveness[45,46], but the effect of activating ERα with cAMP on invasion is not clear. We used a native-like

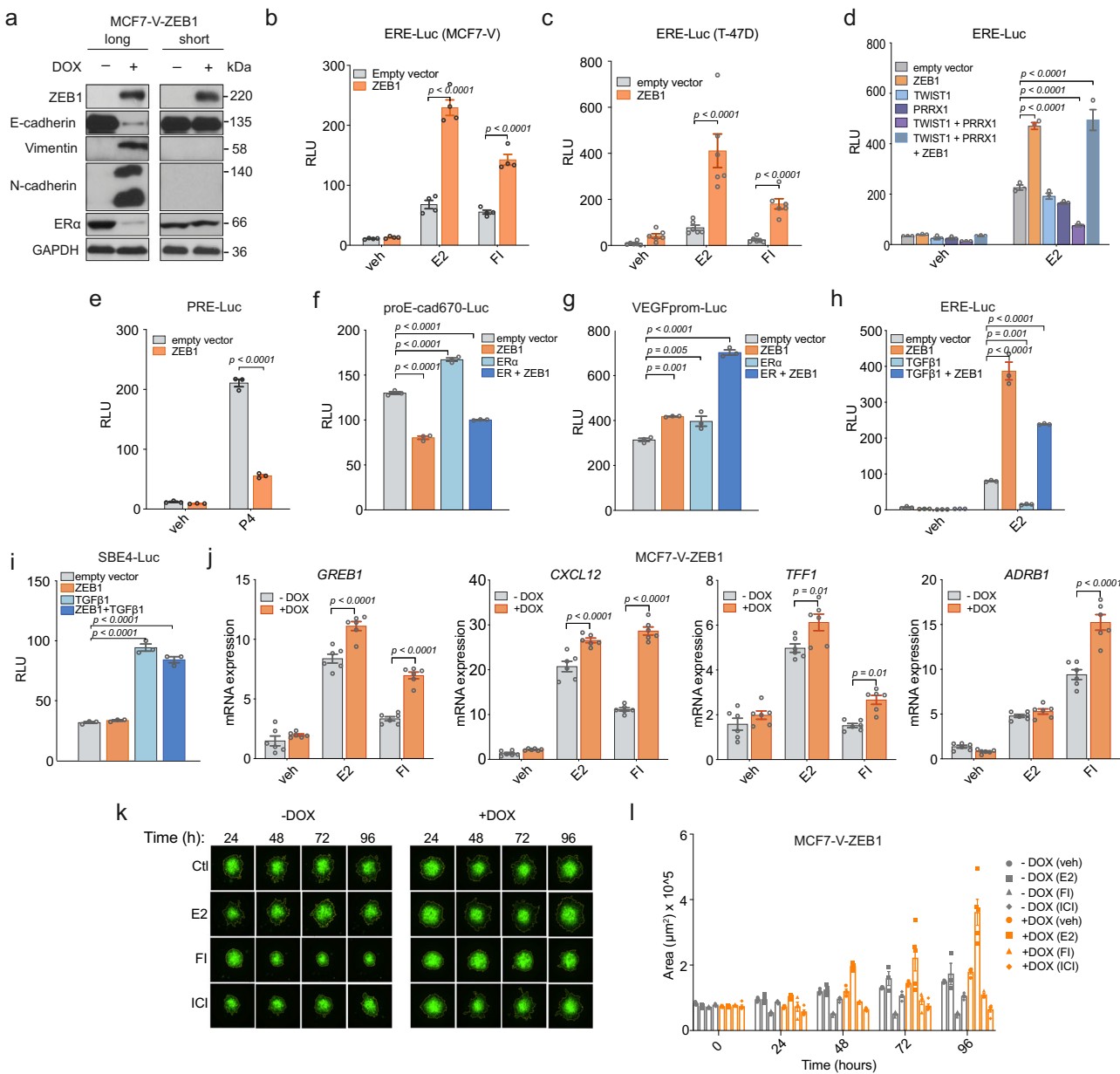

**Fig. 1 Expression of ZEB1 potentiates ERα activity and invasion of breast cancer cells. a** MCF7-V-ZEB1 cells in the presence of doxycycline (+DOX) express ZEB1. Immunoblots show the levels of EMT markers and ERα after long-term (8–12 weeks) and short-term (1–2 weeks) expression of ZEB1 (results are representative of $n = 3$ independent experiments). **b–i** Luciferase reporter assays with transiently transfected cells as indicated, including for ZEB1, which was expressed from plasmid pTRIPz-puro-HA-ZEB1 with DOX treatment. Except for the experiments of **b**, **c**, all assays were done with HEK293T cells. The activities of ERα and PR, and the E-cadherin, VEGF, and TGFβ-responsive promoter activities were monitored with the reporter plasmids ERE-Luc, PRE-Luc, and proE-cad670-Luc, VEGFprom-Luc, and SBE4-Luc, respectively. The luciferase activities (RLU) are expressed relative to the activities of the internal transfection standard, Renilla luciferase. Graphs are based on $n = 4$ for (**b**), $n = 6$ for (**c**), and $n = 3$ for (**d–i**) biologically independent experiments. **j** Expression of ERα target genes in MCF7-V-ZEB1 cells; mRNA levels were analyzed by RT-qPCR following 6 h of treatments as indicated; $n = 3$ biologically independent experiments, each including $n = 2$ technical replicates. **k** Representative images of a three-dimensional (3D) tumor invasion assay with MCF7-V-ZEB1 cells. **l** Invasion kinetics based on the area of $n = 2$ independent spheroids examined over three independent experiments as shown in **k**. veh, E2, FI, and ICI stand for vehicle, 17β-estradiol, forskolin + IBMX, and fulvestrant, respectively. All error bars represent standard errors of the means (mean ± SEM). In **b–j**, $p$ values are indicated above the bars; statistical significance was determined with one-way ANOVA for **f**, **g**, **i** and a two-way ANOVA for all other panels. Source data are provided as a Source Data file.

three-dimensional (3D) tumor microenvironment model[47]. Notably, in the presence of E2, tumor spheroids expressing ZEB1 (+DOX) grew to a larger size within 96 hours (h) of embedding in collagen compared to control spheroids (−DOX) and displayed a significantly increased dissemination from the main spheroid body into the surrounding matrix (Fig. 1k, l). cAMP/PKA signaling was induced using FI. In accordance with recent

findings that PKA activation reverses the EMT and induces a mesenchymal-epithelial transition (MET)[48], FI suppressed the invasion of cells. Moreover, ZEB1-expressing cells preceded the −DOX cells in the invasion, but the antiestrogen ICI suppressed the invasion (Fig. 1k, l). These observations suggest that, in response to E2, ZEB1 enhances ERα-mediated cell invasion. 3D invasion assays with T47-D cells showed similar results

(Supplementary Fig. 1i). We used the Gene expression-based Outcome for Breast cancer Online (GOBO) tool to correlate ZEB1 expression with outcome in ERα+ and ERα− breast cancer patients. Interestingly, higher levels of ZEB1 improve overall survival (OS) and distant metastasis-free survival (DMFS) of ERα+ patients (Supplementary Fig. 1j, k). In contrast, ERα− patients showed no correlation with OS, but higher ZEB1 expression adversely affected the DMFS (Supplementary Fig. 1l, m). This indicates that with relatively high levels of ZEB1, the ERα status determines the outcome.

**ZEB1 induces transitional sensitivity to ERα antagonist**. We determined with the GOBO tool whether the outcome in breast cancer patients treated with the ERα antagonist tamoxifen correlates with ZEB1 expression levels. Indeed, improved relapse-free survival (RFS) and DMFS in tamoxifen-treated patients with higher expression of ZEB1 suggested that an increased ERα activation due to ZEB1 could induce a transitional sensitivity to 4-OHT during early EMT stages (Supplementary Fig. 1n, o). We performed a cell cycle assay with increasing concentrations of the active tamoxifen metabolite 4-hydroxytamoxifen (4-OHT). ZEB1 appeared to sensitize the cells to 4-OHT as indicated by the observation that ~75% of the cells were arrested in the G0/G1 cell cycle phase, compared to only ~60% for control cells (Supplementary Fig. 1p).

**ZEB1 and ERα form a transcriptional complex**. Our data suggest that, during early/hybrid EMT states, ZEB1 can functionally modulate ERα responses. To determine the underlying mechanisms, we tested whether ZEB1 affects the recruitment of ERα to its chromatin binding sites. ZEB1 increased ERα recruitment to the known enhancers of the ERα target genes *GREB1* and *TFF1*. Without activation of ERα, ZEB1 could not increase ERα recruitment (Fig. 2a and Supplementary Fig. 2a). We performed ERα chromatin immunoprecipitation-sequencing (ChIP-seq) with MCF7-V-ZEB1 cells induced to express ZEB1 for 1 week to determine whether there were any global changes induced by ZEB1. The hierarchical clustering of ERα ChIP-seq signals revealed global similarity between replicates (Supplementary Fig. 2b)[37]. Using a differential binding analysis, we compared these results to our previously reported ERα ChIP-seq data for the MCF7-V parent cells. We uncovered 3149 new ERα-binding sites (ERBSs) induced by E2 and 2156 ERBSs induced by FI, all unlocked by ZEB1 (Fig. 2b, c and Supplementary Data 1). The Genomic Regions Enrichment of Annotations Tool (GREAT) for the functional annotation of ZEB1-induced ERBSs revealed biological functions predominantly related to EMT, migration, and activation of WNT signaling (Fig. 2d, e). GREAT predicted several phenotypes associated with abnormal bone morphogenesis, being in line with the proposed functions of ZEB1 during bone development and osteoblast differentiation (Supplementary Fig. 2c, d)[23]. Interestingly, with a de novo motif analysis, we found the ZEB1 motif to be enriched in a subset of ERBSs (Fig. 2f and Supplementary Data 2). We selected ERBSs that were induced by either E2 or FI and unlocked by ZEB1 for further analysis. Genome browser views of the E2-induced ERBSs associated with the genes *TBX2* and *ANXA3* and of the FI-induced sites associated with *CEP89* and *SLC25A24* showed significantly increased ERα recruitment upon ZEB1 expression (Supplementary Fig. 2e and Supplementary Data 1). These could all be verified by ChIP-qPCR (Fig. 2g, h). We then wondered whether ZEB1 might also be recruited to ERBSs. This appears to be the case at least for some sites since ZEB1, for which we used a known site in the *LAMC2* promoter (−96 bp) as a positive control, significantly bound the *GREB1* (+5 kb) and *TFF1* (+0.5 kb)

ERBSs, but apparently not some more remote *TFF1* regions (Fig. 2i).

We extended our findings by comparing ZEB1-binding sites and ERBSs on a genome-wide scale. We performed a ZEB1 ChIP-seq experiment with MCF7-V-ZEB1 cells induced for ZEB1 expression for 1 week. Peak calling revealed 37,922 binding sites for ZEB1. These were compared with the 36,292 E2- and 25,539 FI-induced ERBSs of the parent MCF7-V cells[37]. The Venn diagram of Fig. 2j highlights the extensive overlap between ERBSs and ZEB1-binding sites, and the genome browser views of Fig. 2k illustrate some individual examples. With a re-ChIP experiment targeting some shared sites (with primers designed for the regions highlighted in Fig. 2k and Supplementary Fig. 2f), we could confirm that the two TFs can be simultaneously present at the same chromatin locations (Fig. 2l); as expected, no re-ChIP signal for ZEB1 could be seen at a site (*GREB1* (−20 kb)) where only ERα binds (Fig. 2k, l). Overall, we find that ZEB1 promotes both the ligand-dependent and -independent recruitment of ERα, and that the two TFs share multiple cis-regulatory regions as part of cooperative transcriptional complexes.

**ZEB1 interacts with ERα, and AP2γ is required for the ZEB1-induced ERα activity**. Our findings suggested that ZEB1 and ERα could be present in the same TF complexes, interacting directly or indirectly. Co-immunoprecipitations (co-IPs) of ZEB1 or ERα confirmed that ZEB1 and ERα are present in the same protein complexes, notably upon activation of ERα with E2 (Fig. 3a). We found that the zinc finger cluster 1 of ZEB1 and the F-domain of ERα are necessary for the interaction and that they may play key roles in mediating the stimulation of ERα activity by ZEB1 (Supplementary Note 1 and Supplementary Fig. 3 in Supplementary Information). However, whether these domains by themselves are sufficient for the interaction, or whether other domains or even other factors are required remains to be investigated.

We performed a motif analysis for the regions present at the intersections of ZEB1 and E2- or FI-induced ERα binding sites (Fig. 2j). As expected, the binding sites of ERα and ZEB1 were among the most significantly enriched motifs (Supplementary Fig. 4a). Remarkably, we found the binding sites of the AP2 family, which includes TFAP2A/B/C, as some of the most highly enriched sequences, ranking higher than motifs for forkhead factors such as the ERα pioneer factor FOXA1 (Supplementary Fig. 4a). We also found the enrichment of AP2 motifs in the ZEB1-induced differentially bound ERBSs (Fig. 2 and Supplementary Data 2). AP2 factors play key roles in regulating differentiation, with the *TFAP2C* gene being expressed in adult mammary myoepithelial cells. AP2γ is involved in breast cancer cell proliferation and metastasis. Moreover, AP2γ is recruited to ERBSs to regulate transcription[49–51]. AP2γ, FOXA1, and ERα jointly target genes of the luminal phenotype during breast cancer progression[52,53]. We decided to characterize the possible involvement of AP2γ and FOXA1 in the ZEB1-ERα transcriptional complex. Intersecting published ChIP-seq data for AP2γ in MCF7 cells[49] with our own data for ZEB1 and ERα, we found 6,019 sites shared among the three factors (Fig. 3b). Overlapping ZEB1 and ERBSs with FOXA1 peaks of MCF7 cells[31] resulted in only 520 shared binding sites (Fig. 3c), suggesting that FOXA1 is not a defining factor for the formation of ZEB1-ERα complexes.

We compared averaged ChIP-seq signal intensities for ZEB1, ERα, AP2γ, and FOXA1 with those of known factors and chromatin marks of estrogen-regulated active enhancers including GATA3, P300, H3K27ac, H3K4me1, and H3K9me3 of MCF7 cells[31]. Compared to FOXA1, AP2γ shows a stronger signal around the center of the common binding sites with ZEB1 and ERα

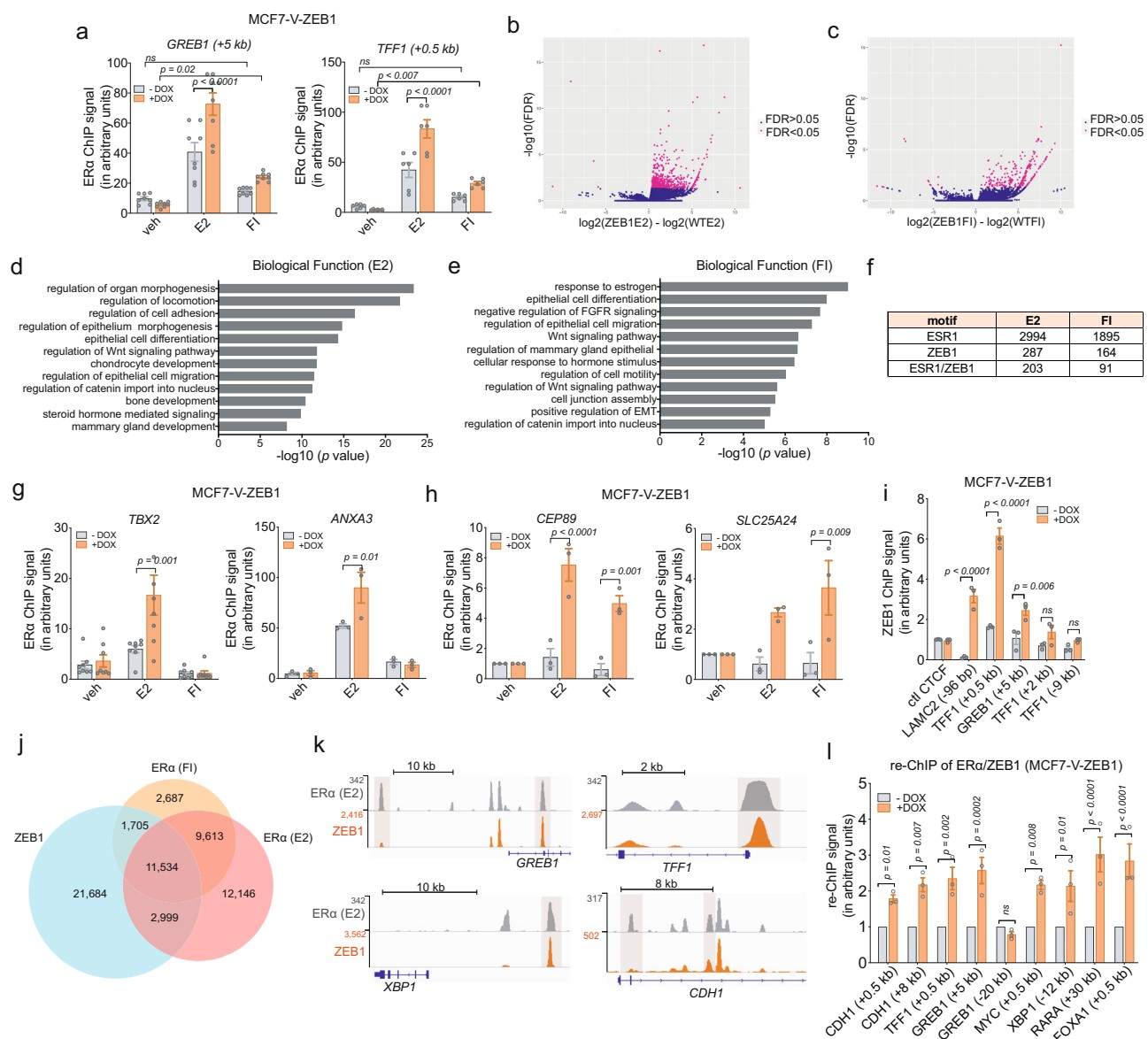

**Fig. 2 ZEB1 enhances ERα recruitment at common binding sites. a** ChIP-qPCR of ERα on the *GREB1* (+5 kb) and *TFF1* (+0.5 kb) binding sites in MCF7-V-ZEB1 cells. ERα ChIP values were normalized to a non-binding region and the input. Recruitment was compared to −DOX with the graph showing the means ± SEM of *n* = 6 biologically independent experiments. **b**, **c** Volcano plots of ERα ChIP-seq with wild-type MCF7-V (data from our previously published data set[37]) and MCF7-V-ZEB1 cells showing the FDR values as a function of the fold-changes of the normalized ERα values of MCF7-V-ZEB1 cells (*n* = 4 biologically independent experiments) compared to the ERα peaks of wild-type MCF7-V cells (*n* = 2 biologically independent experiments) treated with E2 (**b**) or FI (**c**). **d**, **e** Functional annotations for the biological functions of E2- or FI-only ERα binding sites, respectively, using GREAT. **f** Table summarizing the number of ERα (ESR1), ZEB1, and ESR1-ZEB1 shared motifs in the ERα ChIP-seq, as found with FIMO (FDR < 0.05; for *p* values, see Supplementary Data 2). **g**, **h** ChIP-qPCR of candidate ERα binding sites from top hits of the ChIP-seq data for E2 (**g**) and FI (**h**). **g** *n* = 4 biologically independent experiments each including *n* = 2 technical replicates for *TBX2*, and *n* = 3 biologically independent experiments for *ANXA3*. In panel **h**, *n* = 3 biologically independent experiments. **i** ZEB1 ChIP-qPCR with MCF7-V-ZEB1 cells with (+DOX) or without (−DOX) ZEB1 (*n* = 3 biologically independent experiments). **j** Venn diagram shows the intersections between ZEB1-binding sites and E2- or FI-induced ERα binding sites from the ChIP-seq data of MCF7-V-ZEB1 (+DOX) and MCF7-V cells, respectively. **k** Genome browser views of ZEB1 and ERα binding sites adjacent to the ERα target genes *GREB1*, *TFF1*, *XBP1*, and *CDH1*. Highlighted sites were analyzed by re-ChIP-qPCR (see next panel). **l** Re-ChIP experiment showing that ZEB1 and ERα co-occupy the indicated shared binding sites (*n* = 3 biologically independent experiments). The *GREB1* (−20 kb) site is a negative control site as highlighted in **k**. veh, E2, and FI stand for vehicle, 17β-estradiol, and forskolin + IBMX, respectively. Error bars represent the standard errors of the means; *p* values are indicated above the bars. Statistical significance was determined with a two-way ANOVA. Source data are provided as a Source Data file and Supplementary Data 1 and 2.

(Fig. 3d), suggesting that ZEB1 and AP2γ can be a part of ERα TF complexes at sites of open chromatin associated with the histone marks H3K27ac and H3K4me1 (Supplementary Fig. 4b, c). Furthermore, ZEB1, ERα, AP2γ, and FOXA1 seem to co-localize to certain binding sites (highlighted regions in Fig. 3e). We

investigated the effects of reducing AP2γ and FOXA1 levels on the ZEB1-stimulated ERα activity. We knocked down AP2γ or FOXA1 expression (Supplementary Fig. 5a) and found that the activation of the ERE-Luc reporter is reduced upon depletion of AP2γ; this reduction is even more prominent in the presence of ZEB1. The

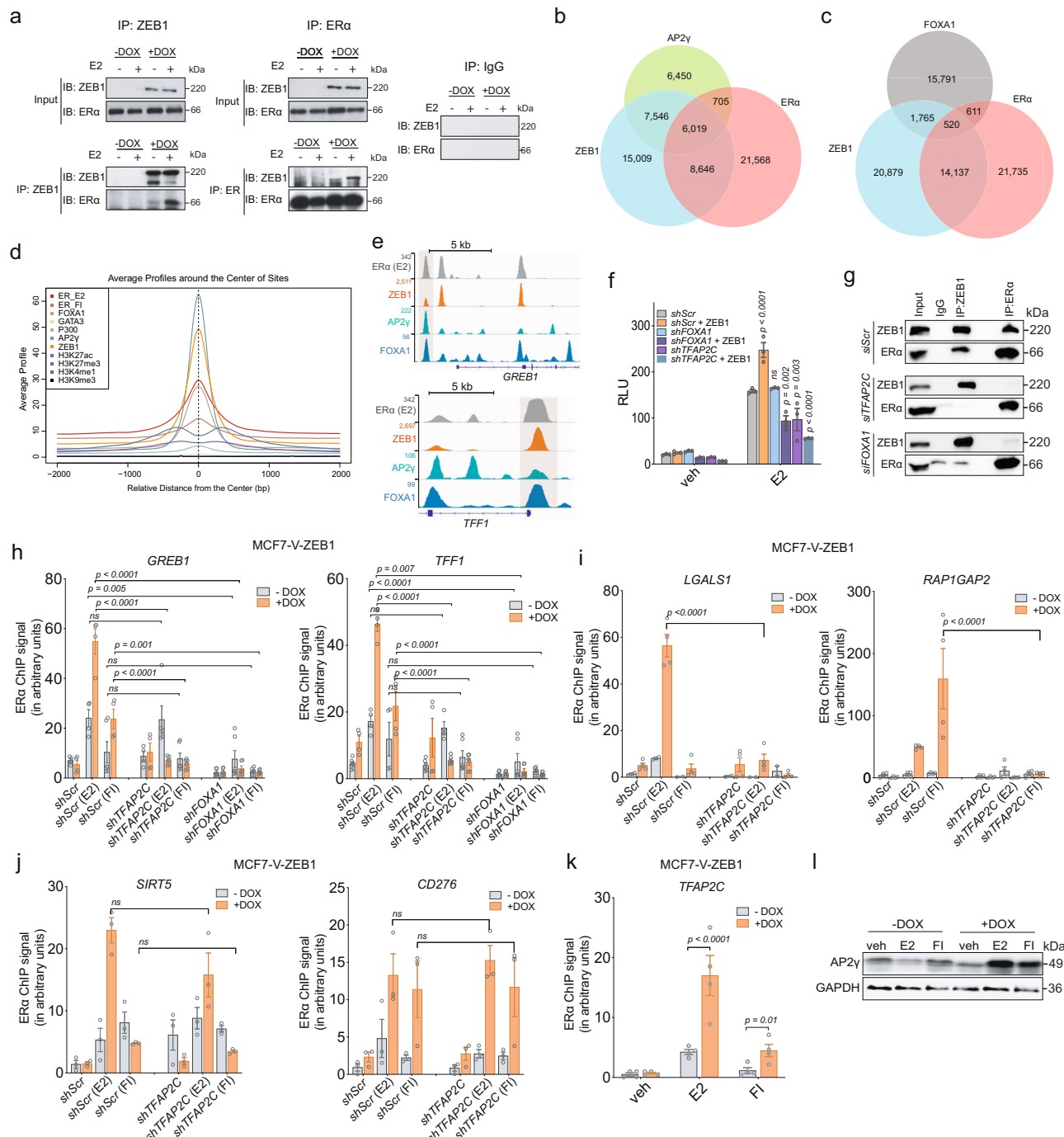

knockdown of FOXA1 did not affect ERα activity by itself in this experimental setup. However, it compromised ERα activity in the presence of ZEB1 even more strongly than without it, and the same could be observed with the knockdown of AP2γ (Fig. 3f).

AP2γ and FOXA1 are known to interact with ERα as part of the same TF complexes[31]. We could confirm by co-IPs from whole-cell extracts that AP2γ and FOXA1 also form complexes with ZEB1, although the interaction between ZEB1 and AP2γ may be more prominent compared to FOXA1 (Supplementary Fig. 5b). We explored whether AP2γ and FOXA1 are required for the physical association of ZEB1 and ERα. For either ZEB1 or ERα, reducing AP2γ and FOXA1 levels (Supplementary Fig. 5c) resulted in the complete loss of the interaction with the other factor in both ZEB1 and ERα IPs (Fig. 3g). We assessed the

consequences of the absence of AP2γ and FOXA1 on the recruitment of ERα. Upon FOXA1 knockdown we observed a similar loss of ERα recruitment to *GREB1* and *TFF1* enhancers independently of ZEB1 expression (Fig. 3h). Remarkably, in the presence of ZEB1, but not in its absence, the loss of AP2γ caused a very significant decrease in ERα recruitment (Fig. 3h). This mirrors the larger impact of the AP2γ knockdown on the ZEB1-stimulated ERα activity in our reporter assays. We tested the impact of an AP2γ knockdown on several ZEB1-dependent ERBSs, identified by our ChIP-seq analysis (Fig. 2 and Supplementary Data 1), in the absence or presence of a FOXA1 binding site (Fig. 3i and Supplementary Fig. 5d–i). Knockdown of AP2γ completely abrogated ZEB1-induced ERα recruitment to an E2-specific site close to the gene *LGALS1*, and an FI-specific site

**Fig. 3 ERα interacts with ZEB1 and requires AP2γ for effective ZEB1-ERα interaction and transcriptional activity. a** Immunoblots of an ERα and ZEB1 co-immunoprecipitation experiment. IPs with extracts from MCF7-V-ZEB1 without or with DOX treatment (for 1 week) were performed with antibodies specific to the exogenously expressed ZEB1 (left) or to the endogenous ERα (right). A control immunoprecipitation was performed in parallel with IgG, blotted, and exposed simultaneously (results are representative of $n = 3$ independent experiments). **b, c** Venn diagrams showing overlap of AP2γ (**b**) or FOXA1(**c**) binding sites with ZEB1 and ERBSs from the ChIP-seq data. **d** Aggregation plot of the binding sites of ZEB1, ERα, AP2γ, FOXA1, GATA3, and P300, and the open chromatin histone marks H3K27ac, H3K4me1, and H3K9me3. Except for ERα, ChIP-seq data were from published data sets: GSE21234 (TFAP2C), GSE25315 (FOXA1), GSE60270 (GATA3, P300, H3K27ac, H3K4me1, and H3K9me3). **e** Genome browser snapshots of ZEB1, ERα, AP2γ, and FOXA1 enhancers of indicated genes. Highlights show shared binding sites for indicated factors. **f** Luciferase reporter assays with ERE-Luc in HEK293T cells infected with lentiviral constructs for shRNAs targeting *FOXA1*, *TFAP2C*, or both mRNAs; scrambled shRNA (shScr) was used as negative control (mean ± SEM, $n = 3$ biologically independent experiments). **g** Co-IPs with HEK293T cells co-transfected with ZEB1 and ERα expression vectors. A control IP was performed with an IgG antibody (results are representative of $n = 3$ independent experiments). **h** Cells infected with viruses for expression of shScr, shTFAP2C, or shFOXA1. ERα ChIP-qPCR values are represented as the fold of the shScr in −DOX (mean ± SEM, $n = 4$ biologically independent experiments). **i–j** ERα ChIP-qPCR of binding sites associated with the genes *LGALS1* and *RAP1GAP2* (**i**), and *SIRT5* and *CD276* (**j**) in MCF7-V-ZEB1 cells infected with shScr or shTFAP2C. **k** ChIP-qPCR of ERBS at the *TFAP2C* 5′-UTR (means ± SEM, $n = 4$ and $n = 3$ biologically independent experiments in **i** and **k**, and **j**, respectively). **l** Immunoblots of extracts from MCF7-V-ZEB1 cells. veh, E2, and FI stand for vehicle, 17β-estradiol, and forskolin + IBMX, respectively. For bar graphs, $p$ values are indicated above the bars; statistical significance was determined with a two-way ANOVA. Source data are provided as a Source Data file.

---

close to the gene *RAP1GAP1*, which both lack a binding site for FOXA1 (Fig. 3i and Supplementary Fig. 5d, e). The binding of FOXA1 and/or yet other factors to the same locations of the *XBP1* and *ANXA2* genes may be responsible for the residual ERα recruitment when AP2γ levels are reduced in cells expressing ZEB1 (Supplementary Fig. 5f, i). We also examined some sites where ZEB1 might bind directly independently of AP2γ (Fig. 3j and Supplementary Fig. 5j). ERα recruitment to these sites remained unchanged in the absence of AP2γ (Figs. 2f and 3j), suggesting a direct association of ZEB1 or the involvement of other pioneer factors. Knowing that ZEB1 could also increase the *TFAP2C* mRNA levels (Supplementary Fig. 5k), we assumed that ERα binding to the *TFAP2C* enhancer should be also enhanced by ZEB1. Our ChIP-seq data in −/+DOX cells demonstrated that ZEB1 enhances ERα recruitment to the 5′-UTR of *TFAP2C* (Supplementary Fig. 5l), which we could confirm by ChIP-qPCR (Fig. 3k). Enhanced binding of ERα to the *TFAP2C* promoter increased the expression levels of the AP2γ protein (Fig. 3l). While ZEB1 activates a distinct ERα response during early EMT, AP2γ binding is necessary for effective and functional ligand-dependent and -independent activation of ZEB1-ERα-bound enhancers.

**ZEB1 reprograms the ERα transcriptome towards a metastatic profile**. To investigate the impact of ZEB1 on ERα-regulated gene expression, we performed RNA-seq of MCF7-V-ZEB1 cells (−/+DOX). Following 1 week of ZEB1 expression, we treated the cells with either vehicle, E2, or FI for 6 h (Fig. 4 and Supplementary Fig. 6a). In the absence of ZEB1 (−DOX), E2 affected the expression of 4046 genes (up/downregulated by at least 1.3-fold; false discovery rate (FDR) ≤ 0.05; Supplementary Data 3), whereas in the presence of ZEB1 (+DOX), 3783 genes were affected (Supplementary Data 4). FI up/downregulated the expression of 8174 genes without ZEB1 (Supplementary Data 3) and 7818 genes upon ZEB1 expression (up/downregulated by at least 1.3-fold; FDR ≤ 0.05; Supplementary Data 4) (Fig. 4a and Supplementary Fig. 6b). In all, 40–50% of these differentially expressed genes were downregulated by at least 1.3-fold (Supplementary Data 3 and 4). ZEB1 caused an overall increase in the expression of ERα target genes (Supplementary Data 4). We performed gene set enrichment analysis (GSEA) followed by clustering the output list of enriched GO terms and generated annotated enrichment maps (Supplementary Fig. 6c–f). ZEB1-shifted the cells towards a more mesenchymal-like and invasive phenotype, with enrichment of genes associated with cell morphogenesis, neuronal differentiation, increased cell motility, and

extensive cytoskeletal changes (Supplementary Fig. 6e, f). We used the ClusterProfiler package in R[54] to classify gene sets with the GO term collections "Biological Process", "Cellular Component", and "Molecular Function". Several functions and processes associated with cell–cell junction, adhesion, and cellular anatomical entity characteristic of the epithelial cell phenotype were inhibited by ZEB1-ERα transcriptional activity (Supplementary Fig. 5g–i).

Most remarkably, using a differential expression analysis, we found that ZEB1 unlocked hundreds of previously undescribed direct or indirect target genes of ERα activated by E2 or FI (Fig. 4b and Supplementary Data 5). The unique GSEA terms from each group showed that these genes are most likely involved in EMT-related phenotypes, stem cell differentiation, bone morphogenesis, and ossification (Fig. 4c–f for unique terms; full GSEA list in Supplementary Fig. 7a–d). Note that the expression of some genes changed in response to ZEB1 expression in the absence of ERα activation (veh) (Fig. 4b and Supplementary Data 5). Gene set analysis with Enrichr (https://maayanlab.cloud/Enrichr/)[55] revealed GO terms such as "negative regulation of cell migration" (Supplementary Fig. 7e). "Early response to estrogen" and "EMT" were among the top terms identified for this set of genes whose expression was altered by ZEB1 in the absence of active ERα (Supplementary Fig. 7f). We also discovered a number of genes that were shared between the ZEB1-unlocked genes revealed by RNA-seq and target genes predicted by a "Binding and Expression Target Analysis" (BETA)[56] of shared ZEB1 and ERα binding sites in our ChIP-seq data (Fig. 2j; Supplementary Fig. 7g, h; Supplementary Data 6). To confirm that the genes unlocked by ZEB1 are ERα targets, we used ICI as ERα antagonist and analyzed mRNA levels from several top hits (Fig. 5 and Supplementary Fig. 8a, b). We selected *DIO2*, *MUC16*, *DSCAM*, and *ESR2*, on the one hand, and *MUC2*, *P2RX7*, *HSPB8*, and *SCG2*, on the other, which are upregulated by ZEB1 in the presence of E2 and FI, respectively (Fig. 5a). *MUC16*, *DSCAM*, and *ESR2* could be confirmed to be E2-induced ERα targets upregulated by ZEB1 (Fig. 5b). Among the FI-induced ERα-dependent targets, *P2RX7* and *HSPB8* were highly upregulated in the presence of ZEB1. The expression profile of *MUC2* was highly unusual in that it was massively induced by ZEB1, but only in the presence of ICI, suggesting that ERα represses this induction under very specific conditions (Fig. 5c).

3D spheroid invasion assays combined with knockdown of newly discovered ERα target genes (Supplementary Fig. 8c) revealed that the depletion of *MUC16*, *DSCAM*, *ESR2*, and *P2RX7* inhibited the invasion of ZEB1-expressing cells, while the

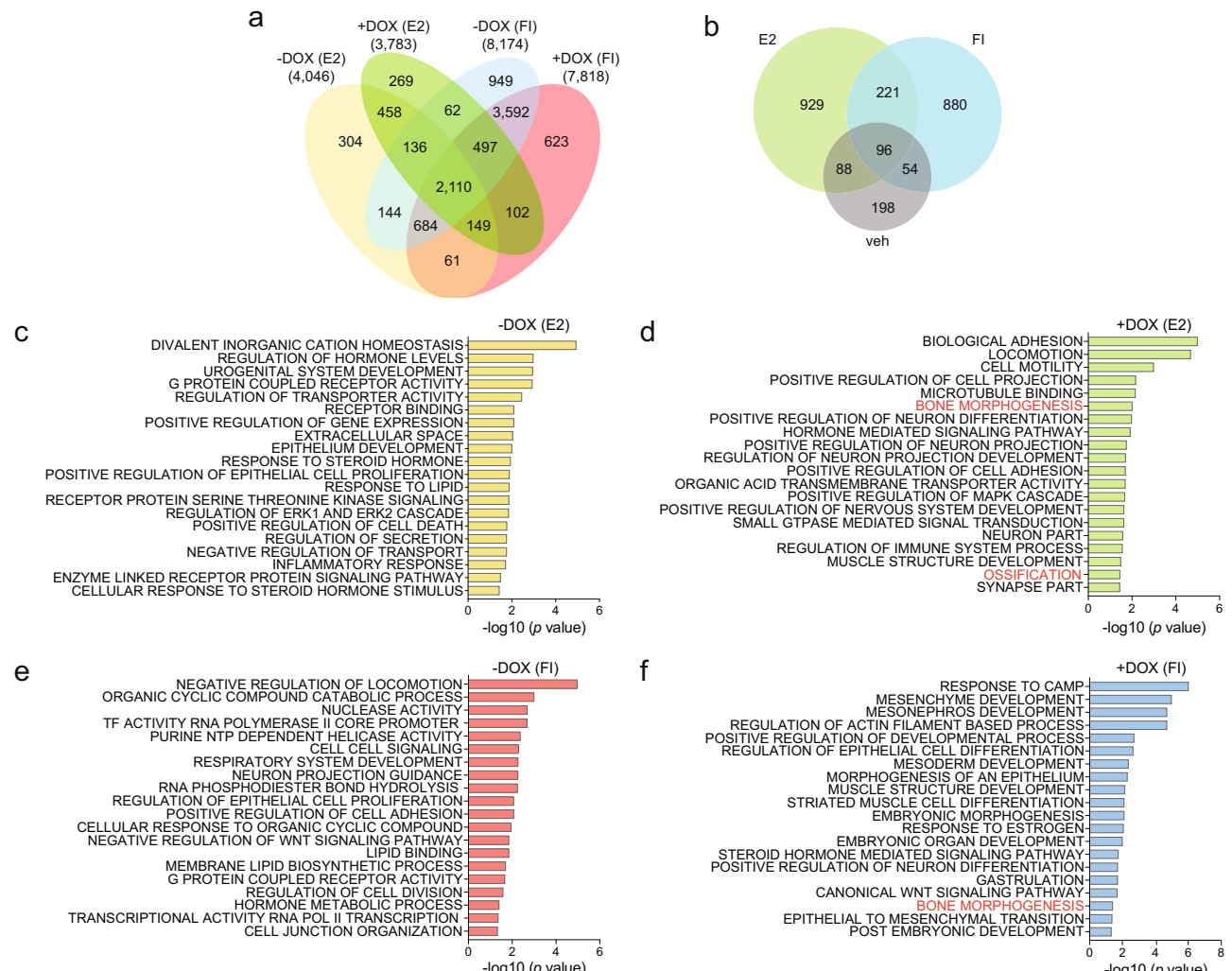

**Fig. 4 ZEB1 reprograms the ERα transcriptome to promote an EMT phenotype of breast cancer cells. a** Venn diagram showing the overlap of the differentially expressed genes for MCF7-V-ZEB1 cells with (+DOX) or without (−DOX) ZEB1 expression in presence of indicated treatments. Fold changes were calculated relative to values from −DOX (veh) ($n = 2$ biologically independent experiments). **b** Venn diagram of the veh, E2- or FI-induced transcriptome changes dependent on ZEB1 expression. **c–f** Bar plots of a GSEA showing the −log10 ($p$ value) of the top 20 unique GO terms associated with significantly altered mRNA expression levels upon E2- or FI-induced ERα activation in cells with (+DOX) or without (−DOX) ZEB1. E2 and FI stand for 17β-estradiol and forskolin + IBMX, respectively. In all panels only genes up/downregulated by at least 1.3-fold were considered for the analysis. Only the genes that had an FDR < 0.05 were included in the analyses. Source data are provided in Supplementary Data 3–5.

invasion capacity of spheroids without ZEB1 expression (−DOX) remained unaffected (Fig. 5d and Supplementary Fig. 8d). *DIO2* depletion had no significant impact on the ability of spheroids to invade in the presence of ZEB1 and E2, corroborating our conclusion that *DIO2* is not a genuine ERα target (Supplementary Fig. 8d). Loss of *MUC2*, *HSPB8*, and *SCG2* completely disrupted the formation of tumor spheroids in cells expressing ZEB1. Cumulatively, these data confirm the pleiotropic role of ZEB1 in modulating the ERα transcriptome and for the acquisition of an invasive cell phenotype during early/partial EMT.

**ZEB1 induces different EMT transition states.** EpCAM has been identified as an epithelial marker for various stages of EMT, and it may play a role in the partially retained epithelial phenotype during partial EMT[57–59]. Moreover, EpCAM^low cells are found to have the greatest number of heterogeneously expressed markers of EMT[17]. We sought to determine how the interaction between ZEB1 and ERα affects gene expression during partial EMT at the single-cell level. To identify the right stage for this experiment, we induced ZEB1 expression in MCF7-V-ZEB1 cells

cultured in a complete medium with a physiological dose of E2 and followed them over time (Supplementary Fig. 9a). We could see a progressive loss of EpCAM, and after >10 weeks of ZEB1 expression, a mesenchymal-like phenotype was evident by visual inspection. At this stage, almost all cells had become EpCAM^low and ERα had disappeared. After 5 weeks, both EpCAM^low and EpCAM^high cell populations were apparent in the fluorescence-activated cell sorter (FACS) profile, and immunoblotting confirmed substantial residual levels of ERα. Therefore, we chose cells at the 5-week time point and performed droplet-mediated single-cell RNA-sequencing (scRNA-seq) of FACS-isolated EpCAM^high and EpCAM^low cells to investigate whether the interaction between ZEB1 and ERα enhances cellular heterogeneity by producing various intermediate EMT stages (for details on scRNA-seq strategy and quality controls, see Supplementary Fig. 9b–d). We identified the highly variable features in both populations including the metastasis-associated genes *MALAT1*, *TFF1,* and *IGFBP5* as top hits (Supplementary Fig. 9e). We discovered 11 distinct subpopulations of cells, including five clusters for EpCAM^high and six for EpCAM^low cells (Fig. 6a). Note that ZEB1

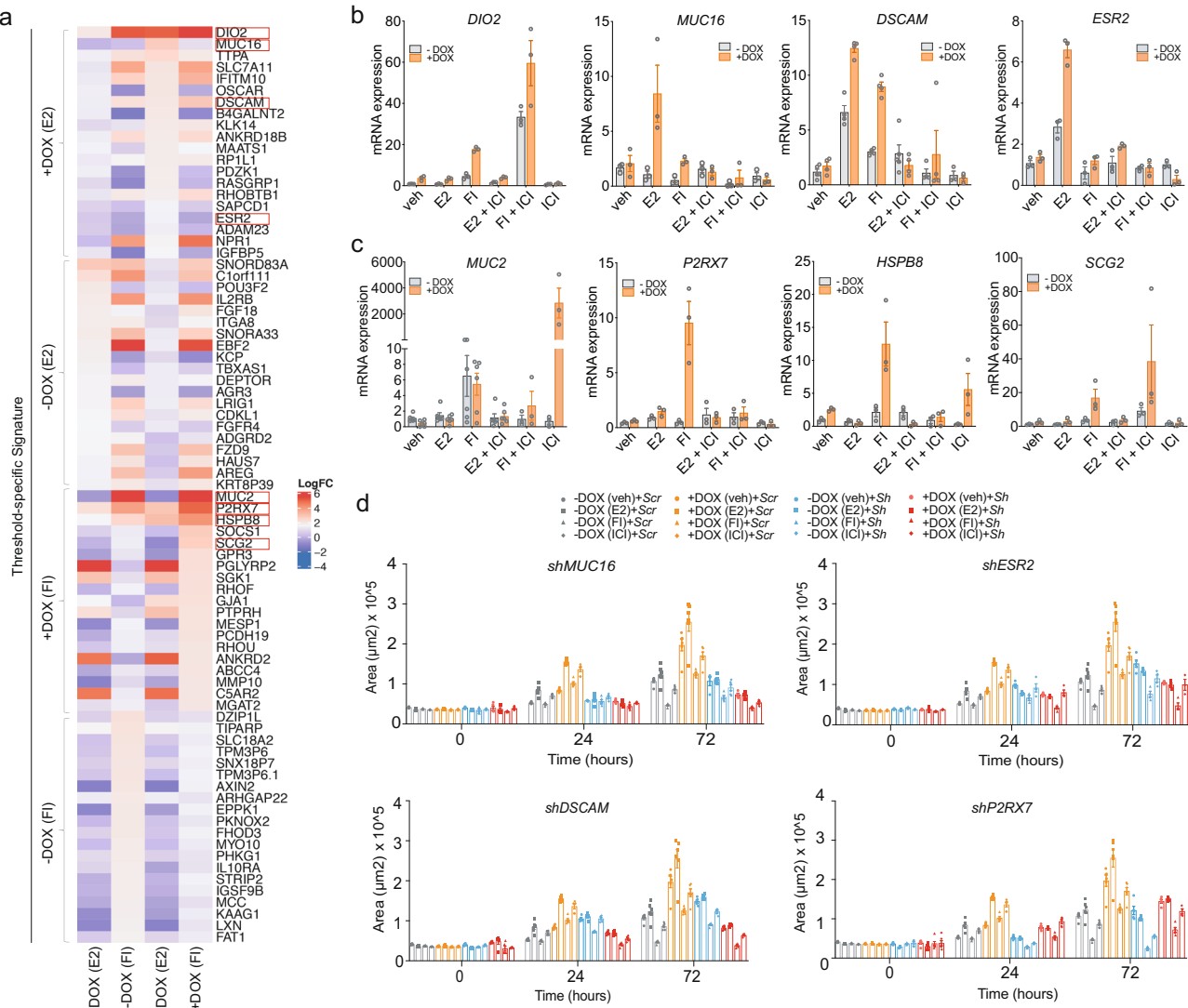

**Fig. 5 ZEB1-induced new ERα target genes are implicated in the invasion. a** Heatmap showing the gene expression levels of the top 20 genes from the RNA-seq data analysis, grouped to highlight treatment-specific signatures (as indicated on the side). Red rectangles around gene names indicate the candidate genes selected for validation and functional assays. **b, c** mRNA levels of candidate genes determined by RT-qPCR (mean ± SEM, n = 3 biologically independent experiments). **d** Bar graphs representing the changes in the 3D invasion ability of spheroids (n = 3 biologically independent samples, each including n = 2 replicate wells) formed from cells with (+DOX) or without (−DOX) ZEB1 expression upon knocking down the expression of *MUC16*, *ESR2*, *DSCAM*, and *P2RX7* with the respective shRNAs. Note that the values for shScr in −DOX/+DOX spheroids (gray and orange, respectively) are from the same data points in all graphs. veh, E2, FI, and ICI stand for vehicle, 17β-estradiol, forskolin + IBMX, and fulvestrant, respectively. Error bars represent the standard errors of the means. All time points are shown in Supplementary Fig. 8d. Source data are provided as a Source Data file.

was expressed in almost all cells of all clusters with the exception of a few cells of clusters 0 and 1 (Supplementary Fig. 9f). Hence, whatever other markers were expressed (see below), ZEB1 was coexpressed with them at the single-cell level. EpCAM^high clusters presented higher expression of *ESR1*, supporting the notion that the progressive loss of the epithelial state is associated with reduced levels of both EpCAM and ERα (Fig. 6b–e). We confirmed the expression of several epithelial and mesenchymal markers at the single-cell level pertaining to epithelial, mesenchymal, and hybrid states as defined by their EpCAM expression (Fig. 6f–h and Supplementary Fig. 9g). This included genes such as *BRIPI*, *ESRP1*, and *CLDN7* as epithelial cell markers[60–63] (Fig. 6f). Interestingly, we found genes such as *ANXA2*, *KRT8*, *HSPB1*, and *TIMP1* to be expressed in both EpCAM^high and EpCAM^low clusters, suggesting that these genes could participate in the establishment of a hybrid EMT state (Fig. 6g). The pioneer factors AP2γ and FOXA1, which are

relevant to ERα activity, are expressed in both EpCAM^high and EpCAM^low clusters with an expression of AP2γ and FOXA1 being higher in EpCAM^low and EpCAM^high clusters, respectively (Supplementary Fig. 9h, i). Genes such as *LGALS1*, *S100A6*, and *LRRC75A* facilitate invasion and were mostly expressed in the EpCAM^low clusters (Fig. 6h). It is noteworthy that *S100A6*, a member of the *S100a* gene family, has previously been linked to hybrid EMT in pancreatic cancer[64].

We used the loss of function experiments with *ANXA2*, *HSPB1*, and *TIMP1* to probe the functional relationship of ZEB1 and ERα with these genes expressed in hybrid EMT cell states. Depletion of each factor (Supplementary Fig. 9j) significantly suppressed cell migration in a wound-healing assay when ZEB1 was expressed and ERα was activated with E2 (left panels in Supplementary Fig. 10a–c). ZEB1 is known to be associated with bone metastasis in invasive ERα− tumors, but which factors cause the early stages of invasion is poorly known. In the context of the functional

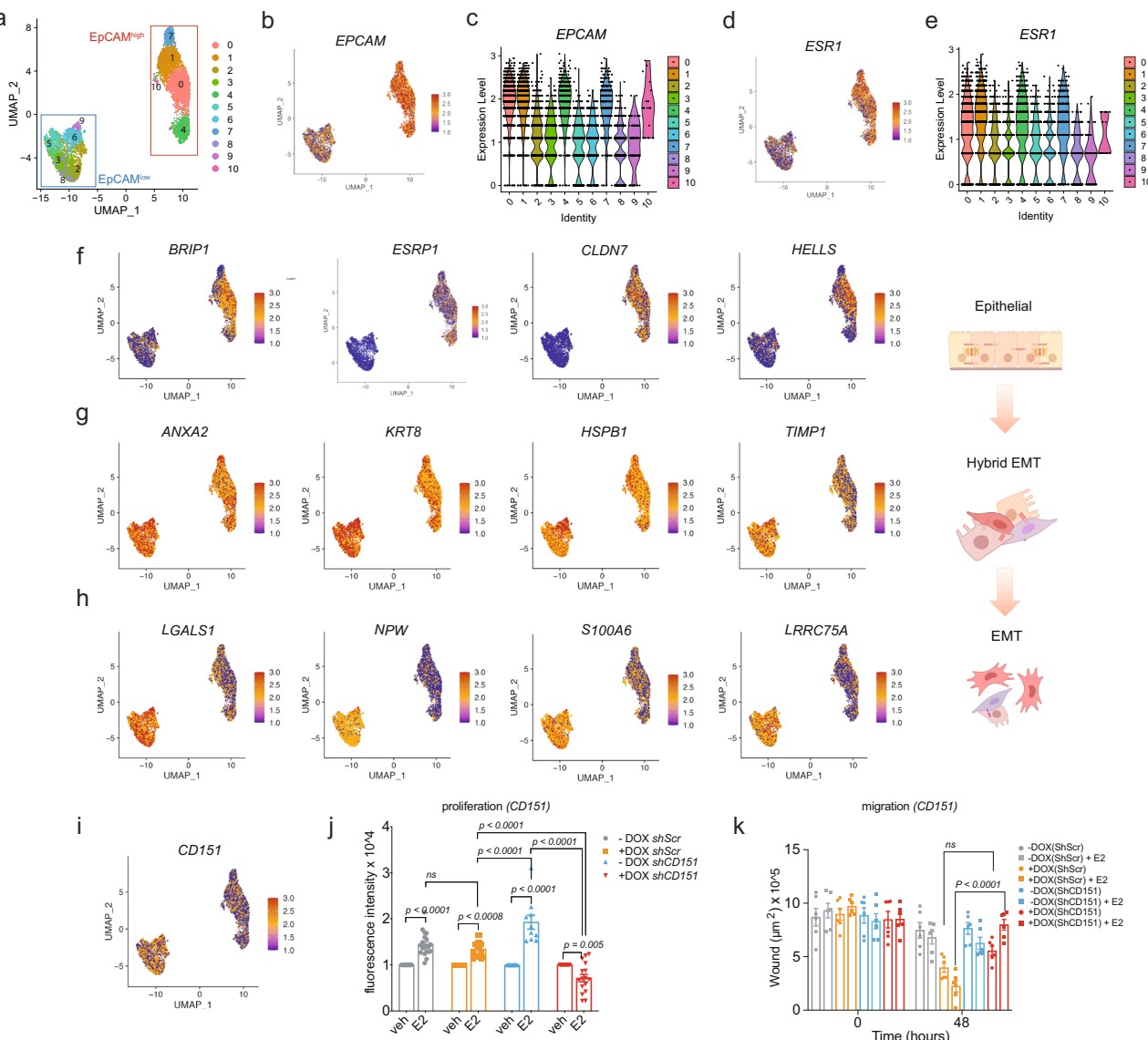

**Fig. 6 ZEB1 induces different EMT states in breast cancer cells with CD151 as a potential marker of early EMT. a–h** show different aspects of scRNA-seq analysis of MCF7-V cells expressing ZEB1 after induction with DOX for 5 weeks to obtain EpCAM$^{high}$ and EpCAM$^{low}$ cell populations. **a** Visualization of the distribution of the merged data sets of both EpCAM$^{high}$ (red rectangle) and EpCAM$^{low}$ (blue rectangle) cell populations, based on the comparison of the transcriptomes of individual cells; the graph was generated by a Uniform Manifold Approximation and Projection (UMAP); the clusters are color-coded and numbered, and each dot represents a single cell. **b** Single cells expressing *EPCAM* in different clusters. **c** Violin plot of the *EPCAM* transcript levels across different clusters; note that the steps at low levels of expression are due to rounding off after normalization. **d** Single cells expressing *ESR1* in different clusters. **e** Violin plot for *ESR1* transcripts with the same color code and numbering as in **a**. **f–h** Expression of top markers of epithelial (**f**), hybrid EMT (**g**), and mesenchymal-like (**h**) states in breast cancer cells upon induction of an EMT by ZEB1 expression. The legend shows a color gradient of normalized expression. The accompanying scheme of the EMT on the right was created with BioRender.com. **i** Expression of the mRNA for the *CD151* cell surface marker across various subpopulations. **j** Relative proliferation of −DOX and +DOX cells with or without *CD151* knockdown exposed to veh or E2 for 72 h (means ± SEM, n = 4 biologically independent experiments averaged from a total of n = 16 technical replicates). Statistical significance was determined with two-way ANOVA. *p* values are indicated above the bars. **k** Quantification of the effect of *CD151* knockdown on the migration of MCF7-V-ZEB1 cells treated with veh or E2 in a wound-healing assay; the Y axis shows the remaining wound area into which cells have not yet migrated (means ± SEM and n = 3 biological replicates each including n = 2 technical replicates). Statistical significance was determined with a two-way ANOVA. *p* values are indicated above the bars. Source data are provided as a Source Data file.

collaboration of ERα and ZEB1, it is striking that cell invasion towards bone in a transwell assay was strongly inhibited by knocking down the expression of *ANXA2*, *HSPB1*, or *TIMP1* specifically in cells expressing ZEB1 and upon activation of ERα with E2. The same knockdowns had no effect on invasion towards lung tissue (middle panels in Supplementary Fig. 10a–c). We, therefore, speculate that *ANXA2*, *HSPB1*, and *TIMP1*, may

be among the genes induced by ZEB1 during early/partial EMT stages that may change the tissue tropism of ERα$^+$ breast cancer cells. When ZEB1 was expressed and ERα activated, reduced *ANXA2* and *HSPB1* expression resulted in a significant reduction of cell proliferation, whereas knocking down *TIMP1* expression increased it (right panels in Supplementary Fig. 10a–c). Therefore, during the early stages of EMT, ZEB1 activation in ERα$^+$

breast cancer cells stimulates the expression of markers of partial EMT, which may impact growth and tissue tropism during the metastatic process.

**Partial induction of EMT by ZEB1 uncovers CD151 as a potential therapeutic target in ERα+ breast cancer cells.** Several EMT stages had been identified in primary mammary tumors based on the expression of the cell surface markers EpCAM, CD61 (encoded by the gene *ITGB3*), CD51 (*ITGAV*), and CD106 (*VCAM1*)[17]. We were unable to detect the expression of these markers in our scRNA-seq analysis of EpCAM$^{high/low}$ cells (Supplementary Fig. 10d), implying the presence of an EpCAM$^{low}$/CD61$^-$/CD51$^-$/CD106$^-$ subpopulation at early/partial EMT stages[17]. CD29 (*ITGB1*) and CD59, among other possible cell surface markers, were expressed uniformly across all subpopulations (Supplementary Fig. 10e), while the tetraspanin CD151 was significantly enriched in the EpCAM$^{low}$/CD61$^-$/CD51$^-$/ CD106$^-$ group (Fig. 6i). The tetraspanin CD151 is a transmembrane integrin involved in metastasis to bone[65–68]. High expression of CD151 supports tumor growth, and this dependency is associated with ZEB1/2[69]. We found ZEB1-binding sites to be associated with the *CD151* gene (Supplementary Fig. 10f), and the CD151 protein levels appeared to be higher in cells expressing ZEB1 (Supplementary Fig. 10g). We wondered whether CD151 might be a therapeutic target during hybrid EMT stages when ZEB1 and ERα are simultaneously expressed in breast cancer cells. We knocked down CD151 (Supplementary Fig. 10g) and assessed cell proliferation in the absence and presence of ZEB1 and active ERα (Fig. 6j). When ERα was activated by E2, CD151 depletion in the absence of ZEB1 boosted cell proliferation. Surprisingly, activation of ERα by E2 upon depletion of CD151 in the presence of ZEB1 resulted in a considerable reduction in cell proliferation (Fig. 6j). Furthermore, in the presence of ZEB1 and active ERα, CD151 depletion dramatically inhibited cell migration (Fig. 6k) and invasion towards bone (Supplementary Fig. 10h). We used the GOBO database and extracted the data for ERα+ breast cancer patients in relation to ZEB1 and CD151 expression levels. Patients with ERα+ tumors that expressed more CD151 had a worse prognosis (Supplementary Fig. 10i), whereas patients with high levels of both ZEB1 and CD151 had a better prognosis (Supplementary Fig. 10j). Overall, these findings suggest that CD151 dependency is associated with the ZEB1+/ERα+ status and that CD151 could be a promising target for inhibiting cell proliferation, migration, and invasion during partial EMT induced by ZEB1 in ERα+ breast cancer.

**ZEB1 reprograms breast cancer cells to promote metastasis in vivo.** High levels of ZEB1 in invasive ERα− tumors are associated with the expression of genes suggested being involved in breast cancer bone metastasis[70]. Our data indicate that ZEB1 modulates gene signatures related to bone development and abnormal phenotypes in ERα+ breast cancer cells (Figs. 2 and 4). This led us to ask whether ZEB1 modifies the organ tropism of ERα+ breast tumor metastases. The conventional MCF7 xenograft models frequently develop metastatic lesions in the lungs, brain, liver, and spleen, but not in bones[71–73]. We, therefore, compared the metastatic potential of control ZsGreen-expressing and ZEB1-expressing wild-type MCF7 cells (+DOX) in xenograft experiments with immunocompromised mice. Since the cells were also marked with luciferase for in vivo detection, we used the bioluminescence (BLI) of the primary tumors and metastatic lesions as a proxy for their relative sizes. Compared to ZsGreen-expressing control cells, ZEB1-expressing cells induced the formation of primary tumors of about the same size (Fig. 7a). All

four mice from the ZEB1 group developed overt bone metastases, whereas in the control group only one out of eight mice showed significant metastatic lesions in bones (Fig. 7b). Considering only the bone metastases, this translates to a higher metastatic burden caused by the ZEB1-expressing cells, as indicated by a significantly higher metastatic index (that is, the ratio of the average bioluminescent radiance of the metastatic organ over that of the corresponding primary tumor) (Fig. 7b). Mice from both groups developed metastases to the lungs (Supplementary Fig. 11a).

There has not been any study on the EMT states of both primary tumors and metastatic lesions for a situation where ZEB1 is expressed. To determine the cellular characteristics within the primary tumors, we performed immunostaining for ZEB1, ERα, EpCAM, and vimentin. Within tumors caused by the ZEB1-expressing MCF7 cells, we could identify different groups of cells with varying degrees of EMT based on the expression of EpCAM and vimentin (Fig. 7c). Group 1 comprises epithelial tumor cells that did not express vimentin. Cells of group 2 solely expressed vimentin and lost cell–cell junctions. Distinct patterns of EpCAM and vimentin coexpression could be seen in cells of groups 3, 4, and 5, indicating partial EMT stages of the tumors (Fig. 7c). Cells of group 3 showed colocalization of EpCAM and vimentin. While cells of both groups 4 and 5 expressed EpCAM, group 4 cells had localized vimentin staining and group 5 cells had diffuse vimentin staining. Furthermore, we observed regions of tumors where EpCAM was present (group 1 in Fig. 7d, e) or absent (group 2 in Fig. 7d, e), regardless of whether ERα or ZEB1 were there (more examples in Supplementary Fig. 11b). Remarkably, ZEB1 tumors contained groups of cells with coexpression of ERα and ZEB1 in the nucleus (Fig. 7f and more examples in Supplementary Fig. 11c). Control tumors were generally in an epithelial state (group 1; Supplementary Fig. 11d), with some regions of tumors showing the expression of both EpCAM and vimentin (group 2; Supplementary Fig. 11d). ZEB1 was not expressed in these tumors and ERα was generally coexpressed with EpCAM (Supplementary Fig. 11e). In metastatic bone lesions from ZEB1 mice, IF staining of ZEB1 and EpCAM demonstrated that cells expressing ZEB1 maintain EpCAM expression (Fig. 7g and Supplementary Fig. 11f). Although ZEB1 staining of metastatic lesions of control mice was largely negative, we did find a small number of cells that expressed low levels of ZEB1 (Fig. 7h). These observations confirm that ZEB1 induces different stages of EMT, that these cells maintain ERα expression, and that an epithelial state persists both in the primary tumor and at metastatic sites.

To further support the tentative conclusion that ZEB1 may redirect metastasis formation to bones, we used two different orthogonal approaches. With an ex vivo bone invasion assay using murine femoral bones, we observed that induction of ZEB1 expression greatly induced the invasion of the ZsGreen-marked MCF7-V cells into the bone when compared to the control cells without ZEB1 (Fig. 7i). In a standard transwell assay, we also readily detected the migration of the ZEB1-expressing cells through the transwell membrane towards bone in the lower chamber, but not to muscle or decellularized bone (Fig. 7j).

## Discussion

The induction of EMT–TFs and the acquisition of mesenchymal characteristics are associated with loss of ERα and resistance to antiestrogen therapies[13,38,39,74], but the impact of early/hybrid states of EMT on ERα signaling had not been investigated. Here, we describe key roles for ZEB1 during early EMT stages in enhancing ERα responses and suggest that the functional ZEB1-ERα interaction may modulate the tissue tropism of breast cancer metastases. Our results reveal that ZEB1 interacts with ERα at shared binding sites at the enhancers of genes involved in EMT,

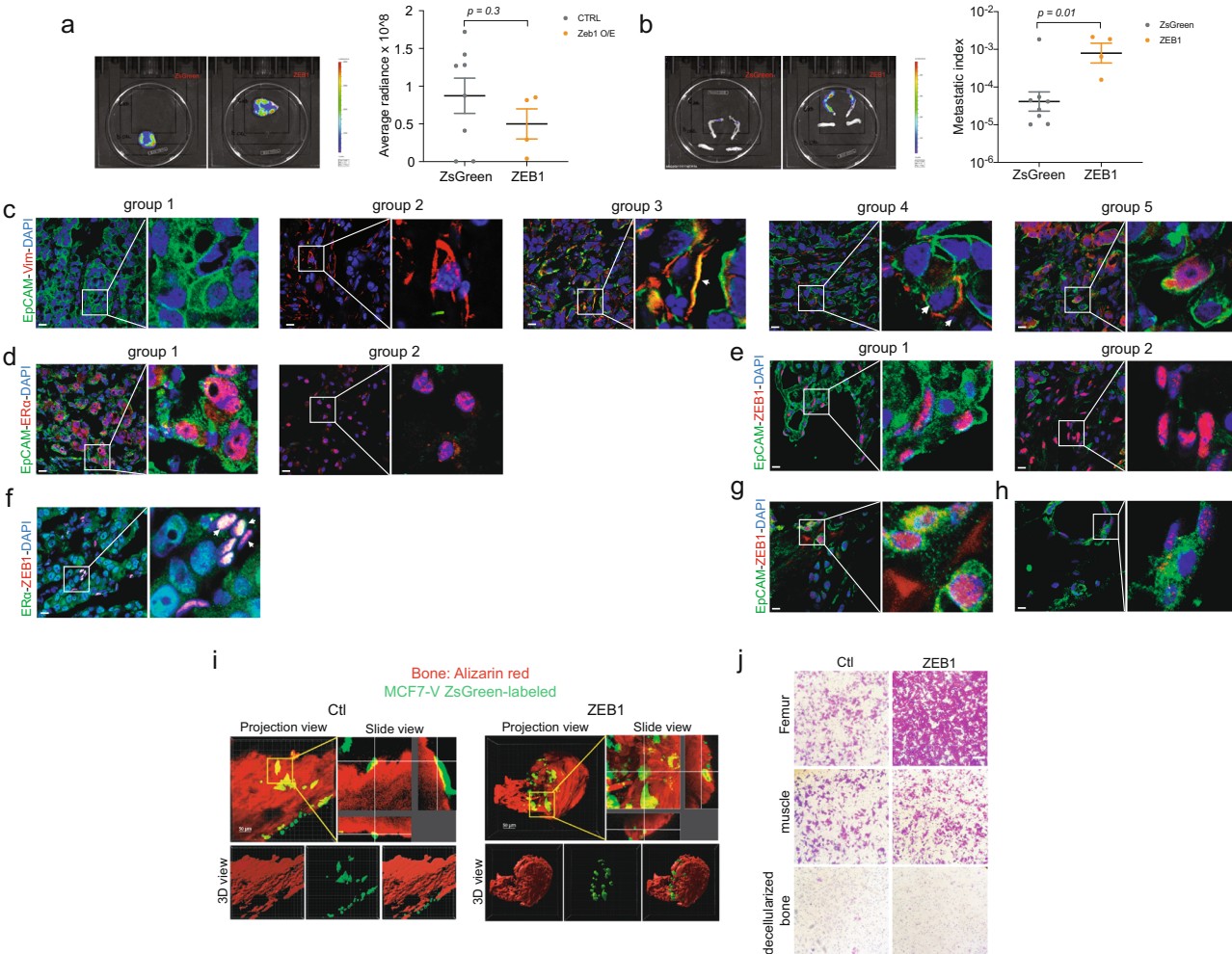

**Fig. 7 ZEB1 promotes bone metastasis in vivo. a** Left panel shows the Bioluminescent images (BLI) of primary tumors of mice injected with one million ZsGreen- or ZEB1-expressing wild-type MCF7 cells into the mammary fat pad. The panel on the right is the corresponding quantitative analysis of the average radiance of multiple primary tumors. An unpaired and two-tailed Student's $t$ test was applied to determine whether tumor sizes are different ($n = 8$ independent animals for ZsGreen cells and $n = 4$ for ZEB1 cells; means ± SEM). **b** Left panel is the BLI images showing metastatic lesions in hindlimb bones of ZsGreen- or ZEB1-expressing wild-type MCF7 cells, respectively. The corresponding panel on the right is a bar graph of the metastatic indices (the ratio of the average radiance of metastases over that of the corresponding primary tumor; see Methods for details) for bones ($n = 8$ independent animals for ZsGreen cells and $n = 4$ for ZEB1 cells; means ± SEM); note that the data are shown after log10 transformation; an unpaired and two-tailed Student's $t$ test was used for the statistical analysis. **c–f** IF images of EpCAM, vimentin, ERα, and ZEB1, and DAPI staining of resected primary tumors formed by MCF7-ZEB1 cells (+DOX) following their orthotopic injection into mouse mammary fat pads (Scale bar = 10 μm). **g, h** IF images of EpCAM and ZEB1, and DAPI staining of metastatic lesions in femurs of mice xenografted with ZEB1-expressing (**g**) and ZsGreen-expressing (**h**) cells (scale bar = 10 μm). **i** Multiphoton confocal microscopy images of an ex vivo bone invasion assay showing the bone explant stained in red with alizarin red and the invading ZsGreen-labeled (green) control (Ctl) or ZEB1-expressing MCF7-V cells. Without ZEB1, the cells remain on top of the bone tissue, whereas ZEB1-expressing cells invade it. The upper left of each panel shows a projection of a Z-stack with 20 images, and in the upper right of each panel a slice of the area delimited by a yellow rectangle. Images were processed and generated with the software Imaris 9.7 to illustrate representative slides and 3D views of bone tissue. Scale bar = 50 μm. **j** Transwell assay to evaluate the migration of ZsGreen-labeled control (Ctl) or ZEB1-expressing MCF7-V cells stimulated by bone and muscle tissues. Crystal violet dye was used to stain the migrated cells. Decellularized bone served as a negative control. Images are representative of three independent experiments. Source data are provided as a Source Data file. Scale bar = 50 μm.

invasion, and bone morphogenesis (Fig. 8). We demonstrate that the interaction between ZEB1 and ERα not only confers augmented transcriptional activation of liganded ERα at the genomic regions that they co-occupy, but also alters the regulation of gene expression by cAMP/PKA-activated unliganded ERα.

Whereas ZEB1 has been known as a repressor for some time, its transcriptional activation function has only been described more recently[22,23]. In line with a previous report[39], we observed that ZEB1 suppresses ERα expression once a mesenchymal-like phenotype is achieved. ZEB1 is known to activate the transcription of TGFβ/BMP pathway genes to support osteoblast

differentiation[23]. In ERα− breast cancers ZEB1 activates the transcription of YAP target genes[22]. We show that ERα protein levels in our luminal breast cancer cells are still relatively high 5 weeks after the induction of ZEB1 expression and that cells maintain epithelial features. Moreover, we could not detect any changes in the number of *ESR1* transcripts by RNA-seq. This suggests that ERα expression could be dynamically modulated at different stages of EMT and that ZEB1 may not have any impact on ERα levels at early/hybrid stages of EMT. The binding of ZEB1 to the two E-box elements within the upstream CpG-rich region of the *ESR1* promoter represses ERα expression[39]. In contrast, our

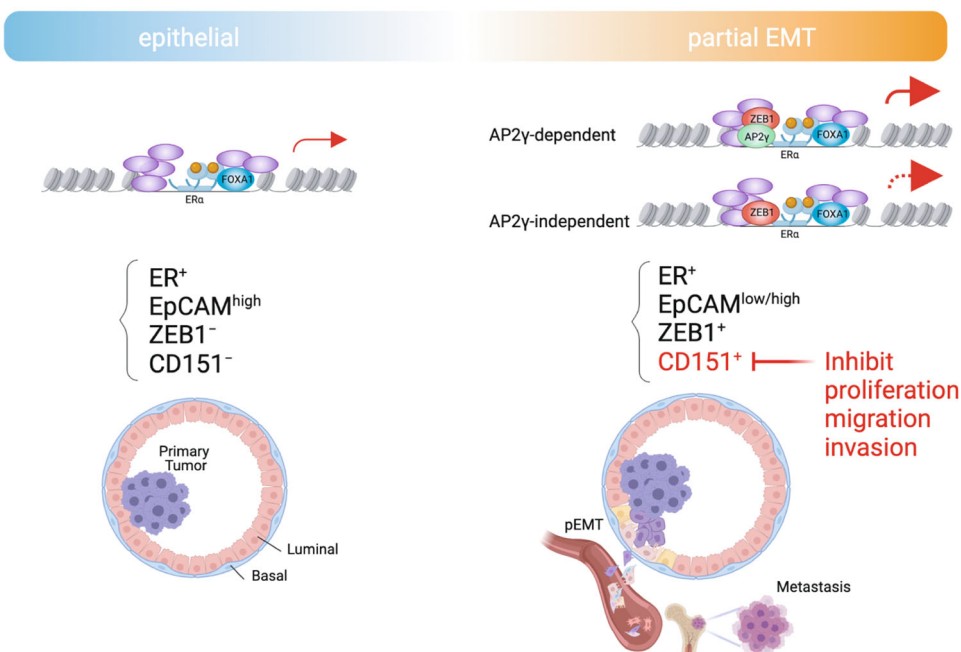

**Fig. 8 Scheme indicating the role of the ZEB1-ERα complex during early/hybrid EMT stages of breast cancer.** Non-invasive primary epithelial breast tumors (EpCAM^high) that express high levels of ERα and are negative for ZEB1 expression are formed by the abnormal proliferation of luminal mammary epithelial cells. FOXA1 acts as the main pioneer factor for the recruitment of ERα for transcription of genes involved in cell proliferation. In early/hybrid states of EMT, AP2γ becomes a determining pioneer factor promoting the formation of a ZEB1-ERα complex at ERα binding sites, which enhances ERα target gene expression. Without AP2γ recruitment, FOXA1 and/or other factors may partially sustain ERα recruitment to ERBSs and ERα-stimulated transcription. This complex reprograms the ERα cistrome and transcriptome towards the activation of genes involved in partial EMT and metastatic dissemination. Expression of specific factors such as CD151 marks the partial EMT state. CD151 could potentially be targeted to prevent cancer cell proliferation, migration, and invasion. The illustration was created with BioRender.com.

ERα reporter construct contains an ERE and its ERα-mediated expression is significantly boosted by ZEB1. We also demonstrate that a *VEGFA* promoter is more strongly activated by both ERα and ZEB1, and that ZEB1 reverses the inhibitory effect of TGFβ on ERα activity. Therefore, we speculate that the existence of an ERE is indispensable for ZEB1 to enhance ERα transcriptional activity. Unlike for ERα⁻ tumors[22,24,39], the presence of ZEB1 in ERα⁺ tumors may improve disease outcomes, perhaps because enhanced ERα activity favors the maintenance of epithelial features and the response to antiestrogens such as tamoxifen.

Our findings support the notion that the pioneer factor AP2γ is involved in the formation of a ZEB1-ERα TF complex. Although AP2γ maintains the mammary epithelial state in ERα⁺ breast cancers[75,76], it can stimulate both EMT and MET by inducing open chromatin states[77–79]. Moreover, we discovered that ZEB1 significantly enhances the ERα recruitment to the AP2γ gene *TFAP2C*, resulting in the upregulation of AP2γ. This suggests a positive feedforward loop where increased AP2γ levels further stimulate the transcriptional activity of the ZEB1-ERα TF complex. Although recent research has shown that ERα regulates *TFAP2C* gene expression[80], we propose a mechanism in which the interplay between ZEB1, ERα, and AP2γ is linked to the regulation of target genes involved in EMT, which may influence prognosis in ERα⁺ breast cancer patients. We discovered that in the absence of AP2γ, ERα recruitment to AP2γ-dependent sites can be partially maintained by FOXA1 (and/or other factors). Interestingly, in addition to the indirect binding of ZEB1 to ERBSs through factors such as AP2γ (Fig. 8), we found that in a partial EMT state ZEB1 directly binds some sites coinciding with an ERBS independently of AP2γ binding. As a result, we propose that ZEB1 engages with ERα at AP2γ-dependent binding sites to activate gene expression (Fig. 8). Whether ZEB1 switches to

acting as a transcriptional repressor upon binding at or near an ERBS in the absence of ERα and AP2γ remains to be investigated. We also uncovered several other motifs, which are associated with ZEB1-ERBSs; implying that yet other TFs may play a role in shaping ZEB1-ERα TF complexes during early EMT states.

Clinical data suggest that cancer cells may start spreading very early during tumorigenesis[81]. Besides, a complete EMT is an extremely rare event in human carcinomas[9]. Tumor cells evolve to different hybrid/partial EMT states, each characterized by different metastatic capabilities[17,82,83]. Strikingly, the circulating tumor cells from breast tumors at different stages show the characteristics of hybrid EMT states and predominantly retain E-cadherin expression[84–86]. It is thus highly possible that each of these states is heterogeneous, and expresses a unique set of markers and different levels of ERα. The ZEB1-ERα cooperative interaction stimulates the transcription of some target genes, which are normally expressed during EMT of embryonic development. We also discovered that the induction of ZEB1 expression leads to the coexistence of subpopulations of epithelial, mesenchymal-like, and hybrid cell states expressing factors associated with cancer cell migration and invasion. Several of these factors are expressed in both EpCAM^high and EpCAM^low cells. For example, annexin A2 (gene *ANXA2*) plays a key role in EMT and metastasis to the bone[87]; keratin 8 (*KRT8*) is a keratin expressed in hybrid EMT states[7]; the small heat shock protein Hsp27 (*HSPB1*) is a regulator of EMT and determining factor in breast cancer stem cells with functions during bone metastasis[88,89]; tissue inhibitor of metalloproteinase-1 (*TIMP1*) can promote EMT, and higher expression of EpCAM and TIMP1 was reported for breast primary tumors, circulating tumor cells, and metastases[90,91]. We discovered that in cells that retain epithelial features and express ZEB1 and ERα, depletion of *ANXA2*,

*HSPB1*, and *TMIP1* resulted in a significant reduction of cell motility and invasion. Therefore, these markers could be gatekeepers of early/hybrid EMT states and potential therapeutic targets.

We discovered and validated the tetraspanin CD151 in the EpCAM^low cell population as a potential target of early/partial EMT stages. In a large cohort of breast cancer patients, elevated CD151 levels were significantly correlated with tumor stage, metastatic potential, and patient survival, and CD151 protein expression was higher in ERα− breast cancers[92,93]. Furthermore, high expression of CD151 is positively correlated with metastasis to bone[68]. We found that a CD151 knockdown significantly inhibited cell proliferation, migration, and invasion in the presence of ZEB1 and active ERα. We propose that CD151 may be a potential therapeutic target for inhibiting cell proliferation, migration, and invasion of ERα+ breast cancer cells during partial EMT caused by ZEB1 (Fig. 8).

Functional annotation of differentially bound ERBSs unlocked by ZEB1 revealed functions associated with EMT, invasion, activation of WNT pathway, and notably bone morphogenesis. This agrees with the established action of ZEB1 during osteoblast differentiation and skeletal morphogenesis[23]. In vitro studies suggest that ZEB1 not only initiates invasion, but that conditioned medium from these invasive ERα− breast cancer cells promotes the maturation of osteoclasts while repressing osteoblast differentiation[94]. In breast cancer patients, the regulation of the BMP pathway by ZEB1 is predicted to correlate with the incidence of metastases in bones, but not in the brain or lungs[70]. We demonstrate with xenograft experiments in mice and ex vivo invasion assays that ZEB1 enhances the invasive and metastatic capacity of ERα+ breast tumor cells and that it may modify the organ tropism of disseminating cells towards bone tissue. Recent studies have reported the coexistence of epithelial and mesenchymal-like cell states, as well as differential hormone receptor expression, within the same tumor cells, demonstrating the occurrence of hybrid EMT states in cancer patients[18,19,95]. Most primary tumor cells maintained the expression of the epithelial markers EpCAM and ERα in our xenograft model while expressing ZEB1 and vimentin. The coexpression of ZEB1 and ERα in breast tumor cells in situ confirms our findings with tissue culture cells regarding the possibility to form a ZEB1-ERα TF complex in cells with partial EMT states. However, the clinical relevance of the ZEB1-ERα target genes in different subtypes of invasive and non-metastatic breast tumors for growth, the formation of distant metastasis, and therapeutic resistance needs to be further investigated.

In conclusion, the present work highlights a mechanism by which ERα signaling is pushed towards activating targets, which shape a phenotype specific to the early stages of EMT and metastasis in breast cancer. The exact components of the TF complex during EMT and the factors responsible for the gradual loss of ERα remain elusive. A small-molecule screen with cells at the early stages of EMT could provide a gateway to developing therapeutic agents targeting the metastatic dissemination at the very early stages.

## Methods

**Antibodies and other reagents**. The anti-ERα rabbit polyclonal antiserum (C1355) (5 µg per ChIP; 1:800 for IF) was from Millipore (Billerica, MA). Rabbit polyclonal antisera against ERα (A300-498A) for immunoblots and co-IPs (1:1000 for immunoblots; 1 µg/mg of protein extract), against ZEB1 (A301-921A) for immunoblots (1:250), for co-IPs (1 µg/mg of protein extract), and for IF (1:400), and against vimentin (A301-620A; 1:500 for immunoblots) were from Bethyl Laboratories. The rabbit polyclonal antiserum against ZEB1 for ChIP experiments and ChIP-seq (10 µg per IP) was from Proteintech (21544-1-AP). Mouse monoclonal antibody against N-cadherin (13A9) (1:1000 for immunoblots) was from Cell Signaling Technology (Beverly, USA). Mouse monoclonal anti-GAPDH (6C5,

ab8245; 1:30,000 for immunoblots) and goat polyclonal antiserum against FOXA1 (1:1000 for immunoblots and 1 µg/mg of proteins for co-IPs) were from Abcam. Mouse monoclonals against AP2γ (6E4/4) and CD151 (H-8) were from Santa Cruz Biotechnology (Santa Cruz, CA, USA) (1:500 for immunoblots). Mouse monoclonal anti-E-cadherin (C36) (1:8000 for immunoblots) and the BV421 mouse anti-CD326 (EpCAM) and IgG1 k isotype control (used for FACS at 1 µg per $3 \times 10^5$ cells) was purchased from BD Transduction Laboratories. For immunofluorescence (IF) experiments the rabbit polyclonal antiserum against vimentin (1:400) was from GeneTex (GTX100619) and the one against EpCAM (1:800) was from Cell Signaling Technology (VU1D9). Alexa Fluor 594-conjugated AffiniPure Fab fragment from goat against rabbit IgG (H + L) (111-587-003) was from Jackson ImmunoResearch Europe Ltd. Alexa Fluor 546-conjugated goat anti-mouse IgG (H + L) (A-11030), Alexa Fluor 488-conjugated goat anti-rabbit IgG (H + L) (A-11034), and Alexa Fluor 488 F(ab')2-goat anti-mouse IgG (H + L) (A-11017) secondary antibodies were from Thermo Scientific. Small interfering RNAs (siRNAs) specific for *TFAP2C* (EHU019581) and *FOXA1* (EHU155811), universal negative control siRNA (SIC007), and the X-tremeGENE siRNA Transfection Reagent were obtained from Sigma-Aldrich (St Louis, USA).

Doxycycline hyclate (Sigma-Aldrich) was used at a concentration of 2 µg/ml for all tissue culture experiments. 17β-estradiol (E2), progesterone (P2), and ICI 182780 were from Sigma-Aldrich. Forskolin, 3-isobutyl-1-methylxanthine (IBMX) and recombinant human TGFβ1 derived from HEK293T cells were from PeproTech (London, UK). Collagen I from rat tail was from Enzo Life Science, UK. Dynabeads-Protein G (10009D, Thermo Scientific) were used for the ChIP experiments (100 µl) and co-IPs (50 µl). A protease inhibitor cocktail (A32965, Thermo Scientific) was used for preparing all cell lysates. Purified rabbit IgG was used as a reference antibody in co-IPs (Sigma-Aldrich, St Louis, USA). HRP-conjugated anti-mouse and anti-rabbit secondary antibodies for immunoblotting were from Agilent Dako (1:8000). 4-hydroxytamoxifen (4-OHT) was from Sigma-Aldrich. AlamarBlue Cell Viability Reagent was obtained from Invitrogen (Thermo Scientific). PEI MAX 40 K (Polysciences) and jetOPTIMUS (Polyplus Transfection) were used for transient transfections. Puromycin was from Cayman Chemical.

**Plasmids**. For ZEB1 expression, the lentiviral doxycycline-inducible construct pTRIPz-puro-HA-ZEB1 was used (a gift from Alain Puisieux's laboratory). Plasmid pTRIPz-puro-HA-ZsGreen was used as a control for pTRIPz-puro-HA-ZEB1. To construct the former plasmid, ZsGreen coding sequences were inserted in the place of those for HA-ZEB1 in plasmid pTRIPz-puro-HA-ZEB1. Plasmid pBABE-puro-mTWIST[96] was from Addgene (ID #1783), and pCMV6-PRRX1 (#RC213276) was purchased from Origene. The plasmid pHAGE-fullEF1a-IZsGreen (plasmid ID 233 from the DNA Resource Core at the Harvard Medical School, Boston) was used to label cells with constitutive ZsGreen expression. We used plasmid HEG0 to express the full-length human ERα[97] and plasmid pSG5-hPR for the expression of human PR[98].

The following luciferase reporters were used: EREtkLuc (XETL)[99] for ERα, PRE-TATA-Luc (a gift from D. McDonnell) for PR, pGL4.10-VEGFprom-Luc (−1000 to −1) for VEGFA (Addgene #66128)[100], proE-cad670-Luc for E-cadherin (Addgene #42083)[101], SBE4-Luc for SMAD (a gift from Bert Vogelstein; Addgene #16495)[102], and the renilla luciferase transfection control reporter pRL-CMV from Promega (E2261).

For knockdowns of *TFAP2C*, *FOXA1*, *MUC16*, *DIO2*, *ESR2*, *P2RX7*, *MMP10*, *SCG2*, *MUC2*, *DSCAM*, *CD151*, *ANXA2*, *HSPB1*, and *TIMP1* the shRNA constructs were generated using the pLKO.1 vector (Open Biosystems) and the target sequences listed in Supplementary Table 1. To produce lentiviruses, the plasmids pMD2G and psPAX2 were used (gifts from Didier Trono's laboratory). The lentiviral UBC-GFP-T2A-Luciferase dual reporter for in vivo imaging and the pMDLg and pRSV-Rev packaging plasmids were from BioCat GmbH.

**Cell culture**. The human breast carcinoma cell lines MCF7 (purchased from the American Tissue Culture Collection (ATCC)) and its variant MCF7-V (see Supplementary Note 1 in Supplementary Information for more details), and human embryonic kidney HEK293T cells (purchased from ATCC) were cultured in Dulbecco's Modified Eagle's Medium (DMEM) complemented with 10% fetal bovine serum (FBS) and 1% penicillin/streptomycin. Human T-47D ductal carcinoma cells were cultured in RPMI-1640 medium supplemented with 0.2 units/ml bovine insulin, FBS to a final concentration of 10% and 1% penicillin/streptomycin. To deprive the cells of steroids, they were cultured for at least 5 days in DMEM without phenol red complemented with 5% charcoal-stripped FBS, 2 mM L-glutamine, and 1% penicillin/streptomycin (hormone-deprived medium). Cells were split with 0.05% (w/v) trypsin in phosphate-buffered saline (PBS), containing 0.02% (w/v) EDTA at least 2× per week. All cells were maintained in 5% $CO_2$ in a humidified incubator at 37 °C. Cells were regularly checked for mycoplasma contamination.

**Virus production and transduction**. HEK293T cells were seeded to a density of $3 \times 10^7$ in a 150 mm dish in standard medium 24 hours (h) before transfection. Lentiviral constructs were co-transfected with plasmids pMD2G and psPAX2. All transfections were performed using the calcium phosphate transfection method. 16 h later, the medium was replaced by a fresh one, and lentivirus/retrovirus-containing

supernatants were collected every 24 h during the next 3 days. Supernatants were filtered and mixed with a 40% sterile polyethylene glycol 8000 (PEG 8000; Sigma-Aldrich) solution by rotating at 4 °C for at least 2 h. The mixes were then centrifuged at 4000 × g at 4 °C for 30 min to pellet the viral particles. Each pellet was then gently dissolved in 1 ml of medium to yield concentrated viral stocks. Cells were infected with concentrated viruses. In all, 24–48 h later, infected cells were selected with puromycin (2 µg/ml for MCF7 and MCF7-V and 3 µg/ml for T-47D cells) for 24 h. Doxycycline-inducible cells were maintained in 2 µg/ml of DOX for the specified duration mentioned in the text.

**Luciferase reporter assays**. 24 h prior to transfections, $6 \times 10^4$ cells for HEK293T and $4 \times 10^4$ cells for MCF7, MCF7-V, MCF7-V-ZEB1, or T-47D cells were seeded in the complete or steroid-deprived medium in each well of a 24-well plate. Cells were transiently transfected with the indicated plasmids and corresponding firefly luciferase reporters and pRL-CMV for renilla luciferase expression using PEI MAX for HEK293T and jetOPTIMUS for other cells. After 8 h, the medium was changed and 24 h after transfection specific treatments were added for 18–24 h including vehicle, E2 (10 nM), FI (10 µM forskolin + 100 µM IBMX), P4 (10 nM), and TGFβ1 (10 ng/ml). Luciferase activity was measured with the dual-luciferase reporter assay (Promega). The firefly and renilla luciferase activities were measured with a bioluminescence plate reader. Normalization to the renilla luciferase internal control was performed to quantify the activity.

**Cell cycle assay**. One day before the treatments cells were seeded at a density of $3 \times 10^5$ per well of six-well plates in a complete medium. The next day, cells were treated with different concentrations of 4-OHT (Sigma-Aldrich) for 72 h. To perform the assay, cells were harvested with trypsin-EDTA and washed with PBS. Cells were fixed in cold 80% ethanol by adding it dropwise to the pellet while vortexing at low speed, followed by incubation on ice for 30 min. After the centrifugation for 10 min at 700 × g, the pellets were washed 2× with cold PBS. Cells were then treated with 100 µl of RNase (100 µg/ml) for 10 min. 300 µl of propidium iodide (50 µg/ml stock) was added to cells to stain the DNA. A Gallios Flow Cytometer (Beckman Coulter) was used to measure the forward scatter and side scatter. Using the FlowJo software, cell debris were gated out, doublets were excluded and the PI histogram plot was applied.

**Protein extraction, co-IPs, and immunoblots**. Cells were washed with tris-buffered saline (TBS), detached with trypsin-EDTA, and harvested by adding a complete medium and centrifuging at 1000 × g for 5 min. The pellets were washed with PBS once and then lysed in ice-cold lysis buffer (10 mM Tris-HCl pH 7.5, 50 mM NaCl, 1 mM EDTA, 10% glycerol, 10 mM Na-molybdate, and 1× protease inhibitor cocktail). Cell suspensions were sonicated for 20 cycles of 20 seconds at high power with the Bioruptor sonicator (Diagenode). After centrifugation at the maximum speed for 5 min, supernatants were collected, and protein amounts were measured with the Bradford assay. For immunoprecipitations, 2 mg of protein extracts were mixed with a specific antibody or control IgG of the same species and incubated overnight at 4 °C on a rotating wheel. On the next day, 50 µl of washed Dynabeads-Protein G were added and incubated for 2 h at 4 °C. Using a magnetic stand, beads were harvested and washed 5x with the lysis buffer supplemented with 0.1% Triton X-100. After the last wash, proteins were eluted from the beads with NuPAGE LDS Sample Buffer (Thermo Scientific) and 10 mM DTT in boiling water for 5 min. To obtain cell extracts for immunoblotting without immunoprecipitation, cell pellets were lysed with lysis buffer supplemented with 0.1% Triton X-100 and protein extracts were mixed with the sample buffer complemented with 10 mM DTT and heated in boiling water for 5 min. Immunoprecipitates and input protein extracts were loaded and separated by SDS-PAGE and transferred to a nitro-cellulose membrane. After blocking the membranes with 5% fat-free milk powder in TBS with 0.2% Tween-20 (TBS-T) for 20 min, specific primary antibodies were added and incubated overnight at 4 °C. Membranes were washed 3× with TBS-T and incubated with a secondary antibody coupled to horseradish peroxidase (Agilent Dako) for 1 h at room temperature. After several washes of the membranes with TBS-T, protein bands were developed and visualized with an ECL kit (Enhanced ChemiLuminescence, Advansta).

**Chromatin immunoprecipitation (ChIP)**. For ERα ChIP experiments, cells grown in a hormone-deprived medium were treated with either vehicle, E2, or FI for 90 min. For ZEB1 ChIP-seq, MCF7-V-ZEB1 cells in a complete medium with a physiological dose of E2 (100 pM) were induced with DOX for 1 week, and triplicate samples with ZEB1 or IgG control antibodies were prepared by pooling five IPs per replicate. ChIP experiments were performed using a previously described protocol[103]. A minimum of $2 \times 10^7$ cells were plated in 150 mm dishes in a specified medium. 100 µl of magnetic beads were washed with 1 ml of blocking solution (0.5% BSA (w/v) in PBS) and incubated with specific antibodies rotating overnight at 4 °C. DNA-protein complexes were crosslinked with 1% formaldehyde for 10 min with gentle swirling. The formaldehyde was quenched by adding 125 mM L-glycine. Cells were rinsed 3× with ice-cold PBS and harvested in lysis buffer 1 (50 mM Hepes-KOH pH 7.5, 140 mM NaCl, 1 mM EDTA, 10% glycerol, 0.5% NP-40 or IGEPAL CA-630, 0.25% Triton X-100) and rocked at 4 °C for 10 min and pelleted at 2000 × g for 5 min at 4 °C. The pellets were resuspended in lysis buffer 2

(10 mM Tris-HCl pH 8.0, 200 mM NaCl, 1 mM EDTA, 0.5 mM EGTA) and rocked gently at 4 °C for 5 min. Nuclei were pelleted by spinning at 2000 × g for 5 min at 4 °C. Each pellet was resuspended in lysis buffer 3 (10 mM Tris-HCl pH 8, 100 mM NaCl, 1 mM EDTA, 0.5 mM EGTA, 0.1% Na-deoxycholate, 0.5% N-lauroylsarcosine, and 1× protease inhibitor cocktail) and subjected to sonication with 30 cycles each 30 seconds at high power. After adding 0.1% Triton X-100, cell debris was discarded by centrifugation at 16,000 × g for 5 min to collect the nuclear extracts.

10 µl of cell lysate was saved as input DNA and stored at −20 °C. The antibody/magnetic bead mix was added to the extracts for each IP and incubated overnight at 4 °C on a rocker. Dynabeads were collected with a magnetic stand and washed 10x with RIPA buffer (50 mM Hepes-KOH pH 7.5, 500 mM LiCl, 1 mM EDTA, 1% NP-40, or IGEPAL CA-630, 0.7% Na-deoxycholate) with the last wash being done with TBS. Inputs and IPs were reverse-crosslinked with the elution buffer (50 mM Tris-HCl pH 8, 10 mM EDTA, and 1% SDS) at 65 °C overnight in a shaker at 700 × g. Dynabeads were discarded and elutions were diluted in TE buffer and incubated with 25 µg/ml RNase for 1 h at 37 °C followed by incubation with 200 µg/ml proteinase K for 2 h at 57 °C. DNA was isolated by extraction with phenol–chloroform–isoamyl alcohol (25:24:1), and the phases were separated with 2 ml Phase Lock Gel Light tubes (5 PRIME) with centrifugation at 10,000 × g for 10 min. Aqueous layers were collected in new tubes with 10 µg glycogen (VWR) and 200 mM NaCl. 100% ethanol was added, and samples were incubated for 1 h at −80 °C and then centrifuged at 16,000 × g for 30 min at 4 °C. Pellets were washed once with 80% ethanol and dried at room temperature. Pellets were resuspended in nuclease-free water. qPCR was performed with the primers listed in Supplementary Table 2. ChIP values were standardized with a non-binding region (the *c-MYC* intron) and normalized with the input values.

**Re-ChIP**. For the Re-ChIP experiments, an ERα ChIP was first performed as described in the previous section, with the exception that after the last wash with RIPA buffer, 25 µl of 10 mM DTT was used to elute the bound chromatin from the beads by incubation at 37 °C with shaking for 30 min[104]. The supernatant was removed and diluted at least 20× with the re-ChIP dilution buffer (1% Triton X-100, 2 mM EDTA, 150 mM NaCl, 20 mM Tris-HCl pH 8.1). The second ChIP was performed with the anti-ZEB1 antibody or control IgG followed by the standard ChIP procedure. All re-ChIP values are relative to the IgG control.

**ChIP-seq and bioinformatic analysis**. For the ChIP DNA library preparation of wild-type MCF7-V and MCF7-V-ZEB1 cells, the Illumina TruSeq protocol was applied for each replicate, and DNA was sequenced using a HiSeq 4000 machine to produce 100 bp paired-end reads. For ZEB1 ChIP-seq experiments with MCF7-V-ZEB1 cells, sequences were aligned to the Human Reference Genome (assembly hg19, NCBI build 37, February 2009) with BWA-MEM (Version 0.7.17)[105]. Peak calling was carried out by using the MACS2 tool (Version 2.1.0) of the Galaxy tool suite (https://usegalaxy.org)[106]. We predicted the fragment sizes generated during the fragmentation step of the library preparation from the alignment results and called the peaks with the input file as control. Only the statistically significant binding sites were kept by R (version 3.6.2) based on the confidence level (−10 × log10 *p* value) of the peak center. Motif analysis was performed with the SeqPos tool using the JASPAR motif matrix. The generation of aggregation plots was done using the cistrome platform (http://cistrome.org/ap/root)[107]. Venn diagrams were produced with the VennDiagram package in R.

For ERα ChIP-seq experiments of MCF7-V-ZEB1 cells, we used four biologically independent replicates for each of the treatment groups. We used the ERα ChIP-seq data from our previously published data set of wild-type MCF7-V cells (GSE109103)[37] to perform a differential binding analysis[37]. FASTQ reads were aligned to the human genome hg19 using BWA-MEM (Version 0.7.17) with standard settings. The quality of the ChIP-seq data was assessed as described in the encode project (https://www.encodeproject.org/data-standards/terms/). MACS2 (Version 2.1.0) with default parameters was used to call peaks on each replicate. These peaks were then used to build a reference data set of binding regions. For each treatment (E2, FI), peaks were added to the reference data set when they were found in at least two replicates. To define standard binding regions each peak summit was extended 50 nucleotides on both sides. This procedure identified 40,720 reference ERBSs for E2 and 26,739 for FI. FeatureCounts (version 2.0.0) was then used to count reads per binding region and generated a count table for E2 and FI treatments. The count tables were then analyzed in R with the edgeR package. Binding regions with a very low number of reads were filtered out (mean of all replicates CPM < 5). The count tables were then normalized, the common dispersion and the tagwise dispersion were estimated with the estimateDisp function. After fitting to a binomial model, the differentially binding sites that were statistically significant were identified with the exact test. For each of the differentially binding regions, sequences of 100 bp surrounding the peak summit were retrieved with the samtools program (version 1.10-3) from the hg19 genome. Identification of specific binding motifs was done with FIMO (version 5.0.5)[108] using the HOCOMOCO v10 collection of TF binding models for human[109] (note that Supplementary Data 5 only includes motifs with FDR < 0.05).

The following publicly available ChIP-seq GEO data sets were reanalyzed and used for comparisons: GSE109103 (ERα)[37], GSE21234 (TFAP2C)[49], GSE25315 (FOXA1)[110], and GSE60270 (GATA3, P300, H3K27ac, H3K4me1, and

H3K9me3)[31]. Sequence Read Archive files from each data set were transferred from the NCBI server and after converting them into FASTQ files, the same procedure as for ChIP-seq analyses mentioned above was followed. The Integrative Genomics Viewer (IGV version 2.8.0) was used to browse and illustrate the binding sites.

**RNA extraction, reverse transcription, and quantitative PCR**. Cells were seeded in the steroid-deprived medium for at least 5 days prior to treatments. RNA was extracted using the guanidium-acid-phenol method from $5 \times 10^5$ cells per well of six-well plates. Briefly, cells were lysed with the TRI reagent (4 M guanidium thiocyanate, 25 mM sodium citrate, 0.5% $N$-lauroylsarcosine, 0.1 M 2-mercaptoethanol, pH 7). 2 M Na-acetate pH 4, aquaphenol and chloroform:isoamyl alcohol (49:1) were added to the cell lysates and mixed vigorously. After centrifugation at $10,000 \times g$ for 20 min at 4 °C, the top phases were collected and RNA was precipitated by adding absolute ethanol and centrifugation at $16,000 \times g$ for 30 min at 4 °C. RNA pellets were washed twice with 70% ethanol and the pellets were dried at room temperature and resuspended in nuclease-free water.

To prepare the samples for RNA-sequencing, the RNeasy Mini Kit and columns (QIAGEN) were used to extract and purify high-quality RNA. A NanoDrop (Thermo Scientific) was used to measure RNA concentrations. Total RNA was reverse transcribed using random primers (Promega) and the GoScript Reverse Transcription System according to the manufacturer's instructions (Promega). qPCR analyses were conducted in 10 µl reaction mixtures including the GoTaq master mix (Promega), cDNA, and specific primer pairs (Supplementary Table 3) with a Biorad CFX96 thermocycler. RNA levels were standardized with the *GAPDH* mRNA as the internal standard control and relative gene expression levels were calculated by the ΔΔCt method.

**RNA-seq**. Before sequencing, the quality of RNA was evaluated with a Qubit 4 Fluorometer (Thermo Scientific). We obtained two biologically independent replicates for each treatment group. The samples were sequenced on a HiSeq 4000 (Illumina). Approximately 50 million paired-end reads of 100 bp per sample were obtained and the quality of reads was checked with the fastqc tool. Sequences were mapped against the Human Reference Genome (assembly hg38, UCSC, August 2015) with the STAR (Version 2.7.0) and count tables were produced with the featureCounts function (Version 2.0.0) in R, containing the number of mapped reads per gene (Supplementary Data 3–5). Differential expression analysis and calculation of fold changes were done with the edgeR Bioconductor package in R[111]. Only genes with a ≥ 1.3-fold change of expression (either up or down) with an FDR < 0.05 were included in the analyses. All Venn diagrams, heat maps, and GO terms were generated in R (version 3.6.2). For the gene set enrichment analysis (GSEA)[112], normalized fold changes from each gene were used from each replicate to generate a table, which was converted to the gct format and used with the GSEA (v4.1.0) software along with the gene ontology gene sets from ontology gene sets of the MSigDB collection, all from the Broad Institute. GSEA tables were then visualized and analyzed with Enrichment Map from Cytoscape version 3.8.2[113] to cluster and annotate the GO interactome.

**FACS and scRNA-seq**. MCF7-V-ZEB1 cells were cultured in a complete medium containing physiological concentrations of E2 (100 pM), and 2 µg/ml DOX to induce ZEB1 expression for the indicated time points. For sorting, cells were harvested with trypsin-EDTA and washed in the FACS solution composed of 1× PBS, 2% FBS, and 1 mM EDTA. $6 \times 10^6$ cells were stained with the BV421-conjugated anti-human CD326 (EpCAM) or the IgG1/k isotype control antibodies for 30 min at 4 °C protected from light. Cells were washed twice with the FACS solution and resuspended in 1 ml of the solution for sorting. The BD FACS Aria III Cell Sorter (BD Biosciences) was used to select living single cells based on the forward and side scatter to separate the doublets and DRAQ7 dye exclusion to exclude dead cells. To perform scRNA-seq of FACS-isolated EpCAM^high and EpCAM^low cells, a minimum of >6000 cells for each group was used. The Chromium Next GEM Single-Cell 3' v3.1 workflow was followed for the library preparation and sequencing was performed with the Chromium Controller system from 10x Genomics.

**Bioinformatic analysis of scRNA-seq data**. Cell Ranger 4.0 (http://10xgenomics.com) was used to process Chromium single-cell 3' RNA-seq output and generate the count table. FASTQ reads were aligned to the reference genome GRCh38-2020-A downloaded from 10X genomics website. All analyses were then carried out in R with the Seurat 3.0 package[114]. QC metrics per cell were calculated with the function PercentageFeatureSet. Only cells with <25% mitochondrial RNA and feature counts between 200 and 7000 were kept. EpCAM^high and EpCAM^low data sets were merged with the function FindIntegrationAnchors, which takes a list of Seurat objects as input and uses these anchors to integrate the two data sets together with IntegrateData. Data were normalized and scaled with the Sctransform function. The percentage of mitochondrial RNA was used as a variable to regress out in a second non-regularized linear regression. The dimensional reduction was performed by PCA and UMAP and the selection of markers specific to clusters of interest was done with the FindConservedMarkers function.

**3D invasion assays**. ZsGreen-labeled cells were seeded in 200 µl of the steroid-deprived medium at a density of 1000 cells per each well of 96-well plates coated with 1.5% of agarose, and incubated for 3 days to allow the formation of compact tumor spheroids[47]. For the assay, the plates were placed on ice and 150 µl of medium was gently removed. Using ice-cold tips, 50 µl of neutralized collagen I (3 mg/ml, at 2× the final concentration) was dispensed to the bottom of each well with six replicates/condition. The plates were centrifuged at $300 \times g$ for 3 min to ensure that the single tumor spheroid of each well is in the center. After 30 min of incubation at 37 °C to allow the collagen I to solidify, 100 µl per well of hormone-deprived medium with 2× concentrations of E2, FI, ICI, and 2 µg/ml of mitomycin to block mitosis was added. Images were acquired using an automated ImageXpress Micro XL confocal microscope (Molecular Devices) every day for 96 h. Images were analyzed with the MetaXpress high-content image acquisition and analysis software.

**IF and confocal microscopy**. To prepare tissues for cryosection, resected primary tumors and bones from mice were fixed with 4% formaldehyde rotating overnight at 4 °C. Tissues were washed twice with PBS, equilibrated first in 20% sucrose in PBS, and then in 40% sucrose in PBS, each for 24 h. Sections from different parts of each tumor and whole bone tissue were mounted in O.C.T. compound (Tissue-Tek), frozen on dry ice, and kept at –80 °C or –20 °C. Primary tumor tissues were cut into 5 µm sections with a cryostat and mounted on Super Frost Plus slides (Thermo Scientific). Slides were dried for 30 min on a slide warmer at 37 °C. For bone a different method of sectioning was used[115]. Briefly, the Norland optical adhesive 63 (Cat#6301, Norland Products, Cranbury, NJ, USA) was applied to the slides. A segment of CryoJane tape (Cat# 39475214) was attached to the trimmed block. 5 µm sections were cut and the tape was placed on a custom-made slide. For the UV curing, the slides were placed on a benchtop UV transilluminator for 10 min to promote adherence of the tissue sections to the slide.

To prepare samples for IF, slides were incubated in a 1:1 solution of methanol and acetone at –20 °C for 20 min. Slides were rehydrated in PBS for 30 min and excess PBS was drained. The edges of each section were marked with a hydrophobic barrier pen. Sections were blocked in blocking buffer (1% goat serum and 1% mouse serum in PBS) for 30 min at room temperature. Slides were incubated overnight at 4 °C with primary antibodies diluted in incubation buffer (1% BSA, 1% goat serum, and 0.3% Triton X-100 in PBS). For the co-staining of ZEB1 and ERα a sequential staining approach was applied using Fab fragments. Slides were washed 3× for 15 min each in PBS, and incubated for 1 h with secondary antibodies and DAPI diluted in incubation buffer, again followed by another three washes for 20 min each in PBS. Slides were then mounted with Fluoromount-G (SouthernBiotech) and visualized using a ZEISS LSM 800 confocal microscope. Images were processed with ImageJ.

**Wound-healing assay**. Cells grown in a hormone-deprived medium were seeded one day prior to the assay in 12-well plates to reach 100% confluence on the day of the experiment. 2 h prior to making the scratches cell were treated with 10 µg/ml of mitomycin c. After being washed with 1 × TBS, two perpendicular scratch wounds were created along the diameter of each well using a 1–10 µl pipette tip. Wells were washed and refilled with fresh hormone-deprived medium with either vehicle or E2. The wound closure area was monitored, and images were acquired every 24 h. The ImageJ software was used to analyze the images.

**Cell proliferation assays**. To measure cell proliferation, cells were seeded at a density of $3 \times 10^3$ cells per well of 96-well plates ($n = 8$), incubated overnight, and then treated with vehicle or 10 nM E2 for 72 h. Cell proliferation was determined using AlamarBlue (Invitrogen) according to the manufacturer's protocol. Staining was measured using a fluorescence-based plate reader with excitation at 560 nm and emission at 590 nm, and the values were normalized to the control samples treated with the corresponding vehicle.

**Transwell invasion assays**. The femur bone, lung, and muscle tissues were dissected from killed mice and cleaned in sterile saline solution. The tissues were sliced into pieces of ~3 × 3 × 3 mm. 20 µl collagen I was pipetted into the center of each well of 24-well transwell plates to immobilize the tissues. Tissue pieces were placed directly into the collagen and 1 ml of DMEM supplemented with 10% FBS, 2% penicillin/streptomycin, and 1% fungizone was gently added on top. Tissues were incubated for 24 h in the incubator to normalize. The medium was carefully replaced with a fresh medium containing 2% FBS or hormone-deprived medium. For the invasion assays of Fig. 7, $1 \times 10^5$ MCF7-V-ZEB1 cells that had already been cultured for 5 days in hormone-deprived medium were seeded into transwell inserts with an 8 µm pore size (Corning) and coated with 20% matrigel (Invitrogen), with 2 µg/ml DOX and either veh or E2. The invasion was assessed 48 h later. For invasion assays of Fig. 8, $1 \times 10^5$ ZsGreen-labeled control (Ctl) MCF7-V or MCF7-V-ZEB1 cells were seeded into matrigel-coated transwell inserts, in medium with 2 µg/ml DOX and 2% FBS. The inserts were incubated for 30 min to allow the cells to settle at the bottom and were placed on top of the tissues. The negative control contained decellularized bone (bone boiled for 10 minutes). In this case, invasion was assessed after 1 week. For crystal violet staining, chambers were rinsed with PBS and the cells inside the transwell inserts were removed using cotton

swabs. Cells on the lower surface of the membrane were fixed in ice-cold methanol for 20 min and stained with 0.5% crystal violet. After washing the stained inserts twice with PBS, the invaded cells were imaged under a light microscope. Bound crystal violet was eluted using 33% acetic acid and absorbance was measured with a plate reader at 590 nm.

**Multiphoton microscopy.** To assess the invasion of cells into the bone, pieces of bone were washed 2× in PBS and fixed in 4% PFA for a minimum of 3 days at 4 °C; PFA was changed daily. Bones were washed and incubated in 2 ml of 0.05% Alizarin Red solution containing 1% KOH and 1 mM HEPES pH = 7.5 gently shaking for 20 min. Pieces of bone were fixed on 35 mm glass-bottom dishes being soaked in 100 μl of 1% low-melting agarose. A SP8 DIVE upright multiphoton confocal microscope (Leica Microsystems) was used to image the invasion of GFP-labeled cells into the bone surface area from 0–200 μm in depth. The Imaris 9.6 software was used to build the 2D and 3D images.

**Xenograft experiments.** Lentiviral particles for labeling wild-type MCF7 cells were generated by co-transfection of the UBC-GFP-T2A-Luciferase dual reporter and the packaging plasmids pMDLg, pMD2G, and pRSV-Rev. Using the same pool of luciferase-labeled cells, lentiviral vectors were used to establish stably transformed cells for DOX-inducible expression of ZEB1 or ZsGreen as a negative control. Cells were cultured with 2 μg/ml of doxycycline for 2 passages before injections. $1 \times 10^6$ cells were injected orthotopically into the mammary gland of NOD/scid GAMMA (NSG) mice, aged 10–12 weeks, and an estrogen pellet (E2-M-17β-estradiol, 60 days, Belma technologies) was implanted subcutaneously into the back of grafted mice. Mice were provided with fresh DOX-containing water (2 mg/ml DOX and 5% sucrose) protected from light. The experiment was terminated at a final stage consistent with the maximum allowed tumor size and burden (maximal approved tumor volume of 2.8 cm³), which was never exceeded. All mouse samples were collected during the same period for each comparative group. All mouse xenograft experiments were carried out in compliance with institutional and cantonal guidelines (approved mouse protocol #3053, cantonal veterinary office of Basel-City). NSG mice were purchased from Jackson Laboratory and kept in pathogen-free conditions specified by the University of Basel and the cantonal veterinary office of Basel-City.

For the bioluminescence (BLI) imaging and quantification mice bearing cells with GFP/Luc were injected intraperitoneally with 3 mg of D-Firefly-Luciferin (Gold Bio, LUCK- 5 G). After 8 min, bioluminescent images of the full mouse were taken using an IVIS Lumina LT (Perkin Elmer). After euthanasia, primary tumor and metastatic organs were imaged separately. Bioluminescence signal analysis was carried out with Living Image, and average BLI radiance was computed as follows: average radiance equals the sum of the radiance from each pixel inside the region of interest divided by the number of pixels in photons/sec/cm² of tissue/sr, where sr = solid angle or steradian. Metastatic indices were calculated as the ratio of the average radiance of the metastatic organ over that of the corresponding primary tumor. To generate the corresponding graphs and statistical analyses, a log10 transformation was applied to the ratios, which normalized the distribution of the data points.

**Statistical analyses.** All experiments were performed at least in three replicate experiments unless mentioned otherwise. All statistical analyses were performed with GraphPad Prism (version 8.3.0) or R. The mean comparison tests were calculated using unpaired and two-tailed Student's $t$ tests, and one-way or two-way analysis of variance. $p$ values ≤0.05 were considered statistically significant. All error bars represent standard errors of the means.

**Reporting summary.** Further information on research design is available in the Nature Research Reporting Summary linked to this article.

## Data availability

The ChIP-seq, RNA-seq, and scRNA-seq data generated in this study have been deposited in the Gene Expression Omnibus (GEO) repository under GEO accession code GSE173562. Source data are provided in this paper.

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

## Acknowledgements

We are grateful to the core facilities of the Faculties of Medicine and Science of the University of Geneva: the Flow Cytometry facility, the Genomics Platform and the Bioimaging Center of iGE3, and the high throughput screening facility (ACCESS Geneva) for their excellent technical support. We thank Dr. Jean-Pierre Aubry and Grégory Schneiter for their generous technical assistance with the FACS procedures. Research in the Aceto lab is supported by the European Research Council (678834 and 840636), the European Union (801159-B2B), the Swiss National Science Foundation (PP0P3_163938, PP00P3_190077, IZLIZ3_182962), the Swiss Cancer League (KFS-3811-02-2016, KLS-4222-08-2017, KLS-4834-08-2019), the Basel Cancer League (KLbB-4173-03-2017, KLbB-4763-02-2019), the two Cantons of Basel through the ETH Zürich (PMB-01-16), the University of Basel and the ETH Zurich, Switzerland. The work of the Picard lab for this study was supported by the Medic Foundation and the Canton de Genève.

## Author contributions

N.M.G. conceived the study, designed and performed most of the experiments, analyzed most of the data, prepared the figures, and wrote the manuscript. M.K.S. conducted the xenograft experiments and bioluminescence analysis. N.H. extensively contributed to the bioinformatics analyses of the ChIP-seq, RNA-seq, and scRNA-seq data. L.B. contributed to experiments with recombinant proteins. N.A. contributed to designing and supervising the xenograft experiments. D.P. supervised and conceived the study, contributed to the design of the experiments, and wrote and critically edited the manuscript. All authors read and approved the final manuscript.

## Competing interests

The authors declare no competing interests.
