## [Peer Review File · Nature Communications]

REVIEWER COMMENTS

Reviewer #1 (Remarks to the Author):

This paper identifies a role for ZEB1 in ER-alpha positive breast cancer, with a detailed characterization of the events in this process. The authors show that ZEB1 increases ER-alpha transcriptional activity in the early stages of metastasis. Links between ZEB1 and known EMT related genes were explored and this was associated with increased cell invasion. Gene regulation of ZEB1 was undertaken and the consequences on ER-alpha DNA binding numbers and target genes were identified, with a large number of changes occurring. A link with AP2-gamma was identified and this functional connection was explored, as was the potential involvement of the pioneering protein FOXA1. Some of the ZEB1-regulated target genes were identified as being functionally important and single cell analysis showed that there were distinct subpopulations of cells when ZEB1 was induced. At the end, the authors assess the involvement of ZEB1 in metastasis and a conclusion is made that ZEB1 is involved in bone metastasis. This is a potentially important topic and the work is of high quality. But the paper feels like lots of small individual parts that have been squeezed together in a paper and there are some fundamental issues with the existing manuscript.

- The paper seems to compare the ZEB1 ChIPSEQ profile from TNBC (ie MDAMB231 & Hs578Ts) with ER-alpha + models. I understand that this is because ZEB1 expression in parental MCF7 cells is probably too low. However, they use ZEB1 ChIPSEQ for all their comparisons (and snapshots) and they compare it with what happens to ER-alpha in MCF7 ZEB1 overexpressing cells. ZEB1 ChIPSEQ in MCF7 ZEB1 overexpressing cells is required otherwise a direct and fair comparison is not achieved. To this point, many of the Venn diagram overlaps are very small and might be what is expected by chance, probably because the authors are comparing two completely different cell contexts.

- The authors conclude based on TF enrichment motifs that AP2G links ZEB1 overexpression to the novel ER-alpha binding sites (and less so, FOXA1). However, in fig 3h, they test the effect of AP2G knockdown at classic ER-alpha targets TFF1 and GREB1. Can the authors use a panel of 5-10 novel ER-alpha binding sites where AP2G motif is present (without FOXA1 motifs) and repeat the experiment to test the impact of AP2G specifically at these "novel" AP2G dependent, ZEB1- ER-alpha sites? Furthermore, the authors suggest that there are some novel ER-alpha binding sites where ZEB1 might bind directly (ie independently of AP2G) but this should be validated.

- Much of the paper relies of luciferase reporter assays which do not possess the biological relevance required.

- The RNA-seq data suggests that a third of the genes in the genome are regulated by ZEB1. Surely this must mean that the threshold is not stringent enough?

- The single cell work is elegant, but I am not sure what can be concluded from it or how it fits in with the rest of the paper. Where is AP2G expressed? Why are the genes identified from the 3D spheroid assays not explored in the different cell populations? At the moment, the single cell analysis only complicates the paper and dilutes the main conclusions.

- The authors tentatively suggest that ZEB1 overexpression redirects metastasis formation to the bone (although bone morphogenesis is not an enriched pathway in figs 4 e+g or fig 7d). However, this is based on 2/5 vs 3/3 with no significant difference in metastatic index. Unfortunately with such small numbers, it is not possible to conclude anything with confidence.

Minor comments:

- The authors use ER-alpha ChIPSEQ for parent MCF7-V cells previously published (Ref #35) while for fig 2j-l, they use for comparison a newly generated ER-alpha ChIPSEQ with wild type MCF7-V cells? What is the difference between these two datasets and why not use the newly generated ER-alpha ChIPSEQ for both comparisons?
- RNASEQ and ChIPSEQ were carried out in biological duplicates. Since this is very few (for RNASEQ, potentially too few) replicates, it would be nice to see how the two replicates cluster together.
- Fig 1l (invasion kinetics quantification): difficult to distinguish between the panels. Also, did they all start from the same position at time 0? This is important to include this, otherwise small differences in starting size/area can be magnified over time, distorting the conclusion.
- Fig 2f (Motifs): are there any p-values associated with this?
- Fig 2m: Re-ChIP: would be good to show re-chip at some sites where ER-alpha binds alone or where ZEB1 binds alone? Or is GREB1 (-2kb) supposed to be one of those sites?
- Fig 5d: the authors state that "the error bars are not shown to simplify the graph". Spheroid type work has a lot of variability – therefore, including the error bars is essential for interpreting this experiment.

Reviewer #2 (Remarks to the Author):

Ghahhari et al analyzed the coregulation of Zeb1 and ERα signaling in metastasis of breast cancer. The authors identified that during early EMT, when the cells are still in an epithelial/partial EMT state, Zeb1 enforces the expression of ERα target genes. Moreover, the two proteins interact and form a complex that involves AP2γ and FOXO1 binding.

Acquisition of endocrine therapy resistance in ER+ breast cancer is a main obstacle leading to rapid progression and metastasis. Therefore a more detailed understanding is crucial and the presented analysis provide important mechanistic insight how tumor cells utilize EMT signals and ER activation for progression, providing sufficient interest for a larger community of Nature communications. The analyses are performed with high technical standards and skills. Although the first part of the manuscript is very straight forward, nicely written, providing crucial analyses to prove the hypothesis and derive solid conclusions, the second part does not follow a comprehensible logic of addressing the next key questions, is less convincing, lacks a thread and needs major revision. Therefore, the manuscript is very difficult to read and to understand. I have outlined my concerns in more detail below:

Figs 1-5 follow a specific logic that falls within a comprehensive scope. The authors identify that

specifically ZEB1 in early pEMT supports ERa activation, whereas it downregulates ERa during later phases of EMT, regulate known and novel common target genes and form protein interaction in a ternary complex with AP2g and/or FOXO1. For the rest of the figures (6-7) the focus on the synergism between ZEB1 and ERa activation is lost. The rationale remains elusive why the authors do not separate cancer cells during ZEB1 induction into EPCAM hi/lo populations for scRNA-Seq. To me, only ZEB1-induced EMT is analyzed in this experiment, independent of ERa activation or expression. Moreover, it was not clearly specified, how identified differentially regulated CD151 (EPCAM hi vs. low) is now connected to the initial findings. In Figure 7 the authors then try to show that ZEB1/ERa activity is changing organ tropism during metastasis based on very unconvincing preliminary data in mouse models (n=3). I suggest to leave the experiments from the last two figures out of the manuscript and focus more on the initial findings. A more thorough analysis of defining the interaction domains in ZEB1 and ERa would be beneficial.

Specific points:

1. Fig. 1 b,c: how was this done exactly? Are these the cells from a, transfected with luciferase constructs only? If not, it would be crucial to test this under DOX conditions for short and long treatment times separately.
2. Fig. 1h: Although the authors show that 293 cells show a moderate activation of the SBE reporter upon TGFbeta treatment, are 293 cells responsive to TGFbeta? Is this the right model to study this? Why not using the Dox-inducible system? 293 cells have a very high endogenous level of ZEB1 without stimulation
3. Fig. 2a and others: authors should provide a negative control showing the lack of enrichment of a non-related genomic region, like of a housekeeping gene or x bp up-/downstream. What does "recruitment over input" mean? The ChIP data should be presented as "percentage of input". It would be very interesting to understand whether also Zeb1 is recruited to these genomic regions in an ERa-activation-dependent manner. Is ZEB1 already present at the target regions in unstimulated, but "+DOX" situation and represses gene expression in the absence of ERa activation?
4. Fig. S3: The interaction between ZEB1 and ERa is convincingly shown, so I wonder why the authors did not follow this path in a more in-depth analysis. I think it would be straightforward to test a construct that contains aa1-272 only (ZF1) in coIP in S3c. Moreover, subdividing ZEB1 into 3 non-overlapping fragments 1-272, 272-882 and 882-1061 would be helpful. In S3e: Why is ctrl+Zeb1 not activating the reporter if E2 was added, like in S3b? S3e: Why was full-length ERa not included in the assay? Why is HBD alone sufficient to activate ERE? Is it binding to the ERE elements in absence of the DNA-binding domain? I don't see why the authors claim that the F-domain should be involved in ZEB1-ERa binding? These experiments need to be more concise or left out because they are inconclusive. Moreover, 293 cells express high levels of Zeb1, so the authors may consider a knockdown of Zeb1 when introducing the ER-domain constructs (S3e). Fig. S3f: input controls of GST/His-tag proteins are missing. Fig. S3a: It needs to be indicated whether the construct "ZF1-ZF2" contains point mutations of the HD or a deletion. In case of the latter, a deletion should be indicated by interrupted/joined parts of the N and C-terminal fragments. Is ZF1-ZF2 the same as dHD? The nomenclature in a and b should be similar! Fig. S3c: I guess the IP was done with anti-HA, so it should be indicated in the figure.
5. Fig. 5d: The authors claim that invasion of ZEB1 expressing cells is inhibited when MUC16, DSCAM etc. are knocked down. This is very unconvincing as only marginal differences are observed and indicating that the cooperation of the two proteins is maybe not acting mainly on invasion. Maybe the data need

to be shown in more simplified manner with less parameters

6. Figure 6/S8: Like I indicated above, I am confused by the twist in the analysis by now dissecting EPCAM hi/lo cells. At which time-point was this done, early vs late DOX treatment and EMT? How does this fit to the initial finding that ER is downregulated in long term Zeb1 expressing cells. How does this relate to ERa? Here only different transition states of EMT induction are analyzed independent of ERa activation. Is there E2 treatment involved? How do the authors come up with focusing on integrins?

7. Fig 6i-m/S8g,h: CD151 appears like a randomly picked target gene and the connection to ZEB1/ERa is enigmatic. Moreover, why is now sensitivity to EGFR inhibition analyzed? The effects in Fig. 6 k,l show that they are likely the result of two independent additive effects, but where is the connection/synergism?

8. Fig. 6m: The Kaplan-Meyer plots need to be evaluated in combination with single effects of Zeb1 hi/lo (Fig. S1) and CD151 hi/lo (not yet shown): is there an improved separation of the two groups/survival benefit? It looks like that mainly Zeb1 high/low is a good prognostic marker independent of CD151.

9. Fig 7: Since the cells are analyzed in vivo for 10 weeks under constant DOX supply, it is difficult to assess whether the effects are due to an early pEMT, when ZEB1 cooperates with ERa or due to a late EMT phase, when ERa is already downregulated. The analysis of tumor size by BLI is not appropriate, especially since the images show that they are of similar size. Since the ZsGreen and ZEB1 cells have been infected separately, the luciferase expression might be at different level. Why was only E2 treatment but no untreated control included? Fig S7f: The conclusion of changes in organ tropism are very unconvincing with the assays presented. A colonization experiment would help to define whether cells injected into the circulation will preferentially colonize the bone or would also create more lung mets when ZEB1 is induced. Again: what is the connection to ERa? In this experiment even no E2 was added, so no ERa activation is involved?

Minor points:

1. "DMFS" is misspelled in the text (DMSF).
2. All Western blots lack Mw standard indicators
3. Fig. S8h: there is a labeling error, as Dox/sh treatments are similar between left and right. Maybe the first 2 lanes are -DOX and the last 2 are +DOX?

Reviewer #3 (Remarks to the Author):

Review NCOMMS-21-17328

In This manuscript, Ghahhari and colleagues investigate the transcriptional cross talk of ZEB1 with Estrogen receptor Alpha (ERa) in the context of hybrid/transitory EMT. The manuscript begins with a series of biochemical approaches to characterize ZEB1 and ERa at the level of transcriptional regulation (figure 1), chromatin regulation (Figure 2) and protein interaction (Figure 3). They then expand this observation using RNAseq (Fig 4-5) and single cell RNAseq (fig 6). Finally, they perform additional in vivo or ex vivo studies to link these observations to phenotypical changes occurring in metastasizing cancer cells. Overall, these observations are extremely interesting and reveal a previously unknown axis of regulation for ERa. The novelty and limitation of this study stems from the transitory nature of these

hybrid phenotypes and their potential role in breast cancer biology. I believe this is potentially an interesting study but I also think that the manuscript requires some work before it could be considered for publication. Some of the analysis are too preliminary and the use and integration of additional datasets from alternative models is not always well justified.

Major Concerns:

- 1- The cross talk between ER and ZEB1 appears to have an “expiration date”, which is represented by the mature EMT transition. The author uses a ZEB1 inducible model. What happens if the DOX is turned off at various stages if ZEB1 induction? When EMT becomes irreversible?
- 2- Following, this, it is essential to understand when and why ZEB1 would be activated in luminal BRCA. The model (figure 8) does not offer any indication, and albeit the authors might think this is not the point of the study, most of the clinical data presented by the same authors suggest that high ZEB1 is good because it reinforces the ERα signalling network. Either the authors focus the paper on the mechanistic aspect, or it should justify this apparent contrast (as ZEB-ER should be short lived and linked to earlier and more productive microdissemination)
- 3- The effect of ZEB1 induction on “invasion” in the 3D models (Fig. 1K,L) should account for potential difference in basal cell proliferation induced by ZEB1 activation (which emerge from figure 1C). Some proliferation 2D/3D measurement would give this reviewer more confidence this is truly invasion and not increase proliferative pressure.
- 4- Figure Supp 1N. Overstatement, these data alone do not support the statement “ZEB1 sensitizes...”, as ZEB1 activation increases the G0/G1 fraction
- 5- Can the author add more details in the text around figure 2? ER ChIP seq was done in MCF7-V-ZEB1. Is this short term, long term induction? Or something else?
- 6- Overlapping ERα from MCF7 ZEB1 (I assume) with ZEB1 from MDAMB231 is not appropriate. This reviewer believes that a ZEB1 ChIP-seq dataset from the same model is required prior to publication (and should be added in the analysis to the current one).
- 7- Page 11 “ We decided.....(Fig 3B). It is not clear from what models the AP2 and FOXA1 dataset come from.
- 8- The FOXA1 KD experiment is very tricky as most cells would die in the process. Indeed, only 50% mRNA suppression was achieved. Hard to interpret.
- 9- Page 12: when the authors refer to “protein complexes” it would be more appropriate to disclose how these experiments were done and then rewrite these sections. Are these interactions occurring in the cytoplasm, nucleus or on the chromatin? Most of FOXA1 is bound to chromatin, AP2 is also considered by some a Pioneer factor, mostly chromatin bound. ER is much more dynamic. What is the authors interpretation of their data? Are these complexes formed in solution or are these data reflecting a more “additive” scenario with FOXA1, AP2 already bound and ER-ZEB1 coming later on in response to ligands or other stimuli?
- 10- Fig 4. More details on how long DOX was given is required for a correct interpretation of the data.
- 11- Page 15. “this suggests the existence....EMT stages”. How robust are these data considering the drop-out effect of scRNAseq? Can the authors show saturation analysis? With scRNA data, often is a question of sensitivity.
- 12- The EGFR inhibitor experiments are vastly exploratory. If EGFR dependency is as short lived as the ZEB1-ER cross talk, when would this vulnerability become valuable (if detectable at all) in a clinical

scenario?

13- The in vivo experiments would require some pathology characterization. For how long was DOX induced here? Why is it that controls n=4 and ZEB n=3? We noted that one of the control in Fig 7B has a striking load to the bone (indeed the pVal is not significant). The discussion of figure 7 should be tone down significantly as the experimental designed is not really conclusive (fig. 7E quantification?)

Overall, I'd be keen to see a vastly reformatted and streamlined manuscript which focus on the main finding without unnecessary deviations or forced clinical relevance.

Minor points

Fig. 1B. The veh results are normalized to 1. It is therefore impossible to judge the effect of ZEB1 induction on basal levels of ERE driven transcription (which is apparent in Fig 1C).

ChIP results from Figure 2D suggest ZEB1 induction as a quantitative effect on ER binding (not an ON/OFF). Suggesting that even in the absence of ZEB1, these loci recruit, in some cell, ER.

Reviewer #4: (Remarks to the Author):

In this manuscript, Nastaran Mohammadi Ghahhari et al. analyzed EMT-related interaction of ER α , ZEB1 and other regulatory factors. They identified and characterized the functional interaction of ZEB1 and ER α by using MCF7 and some cell line systems. They also conducted ChIP-seq analysis and invasion assays for confirming transcriptional programs and EMT phenotypes. They found that ZEB1 modulate ER α signaling by recruiting novel ERBSs and sharing the regulatory elements. The authors additionally performed scRNA-seq and identified CD151 as therapeutic targets during early/hybrid EMT stage. They finally conducted xenograft/ex vivo assays to unveil relationship between the ZEB1-associated program and bone metastasis. The authors showed a lot of results using MCF7-based systems. However, they should clearly describe whether the obtained results could be represented in real cancer tissues. There are some points that need to be addressed to improve the manuscript as below;

Points;

1. The authors need to explain the detailed experimental condition, strategy and purpose of scRNA-seq. In scRNA-seq analysis, they should indicate which cells come from EPCAM high or EPCAM low fractions of FACS sorting and show expression levels of ZEB1 in each cell. I'm not sure which time course samples were used for scRNA-seq experiments. I also think that bulk RNA-seq analysis is enough for the comparison of the two populations according to EPCAM expression.
2. Are cells at the hybrid EMT state in real clinical samples of breast cancers? The authors should add information of the obtained results in clinical samples.

Minor points;

1. In Fig. 6a, it is not easy to associate colors and cluster numbers. To easily understand Figs. 6c and e, the authors should add the cluster numbers to the UMAP or some description to the figures.
2. They defined MKI67 as epithelial cell markers in scRNA-seq analysis. To my knowledge, MKI67 is a proliferation marker and their expression patterns are dependent on cell cycle state. The authors should describe the association between the findings of scRNA-seq and cell cycle state of each cluster if they did not perform regression of cell cycle effects.
3. In p. 13, please specify the number "> 1,000".
4. In Fig. 8, the authors represented the overview of the obtained results. I recommend that they add information of EMT stage (early, hybrid and late) and other key molecules which were identified in this study, such as CD151 if possible.

Point-by-point response to reviewers' comments

Note that our responses to the reviewers' comments are in blue.

Building on very constructive comments, we performed several additional experiments and substantially revised the manuscript to accommodate those new results and all other edits required to address the reviewers' comments. The following are the major changes in brief:

- We did the ZEB1 ChIP-seq.
- We strengthened the xenograft data.
- Impact of ZEB1 on proliferation: We did the experiment as suggested.
- Pathology: We performed an immunofluorescence analysis of various markers for both primary tumors and bone metastases.
- scRNA-seq: We did additional experiments focusing on several individual genes to connect the scRNA-seq data to the other parts of the story, and we revised the text to clarify how this all fits together.
- Clinical relevance: We had addressed this from the outset, in Supplementary Fig. 1, but we also added more references for the relevance of hybrid EMT states in the Introduction.
- Please note that we slightly modified the Title of the manuscript to match the fact that we can now state with confidence that ZEB1 alters the tissue tropism of breast cancer metastasis towards bone.

Reviewer #1 (Remarks to the Author):

This paper identifies a role for ZEB1 in ER-alpha positive breast cancer, with a detailed characterization of the events in this process. The authors show that ZEB1 increases ER-alpha transcriptional activity in the early stages of metastasis. Links between ZEB1 and known EMT related genes were explored and this was associated with increased cell invasion. Gene regulation of ZEB1 was undertaken and the consequences on ER-alpha DNA binding numbers and target genes were identified, with a large number of changes occurring. A link with AP2-gamma was identified and this functional connection was explored, as was the potential involvement of the pioneering protein FOXA1. Some of the ZEB1-regulated target genes were identified as being functionally important and single cell analysis showed that there were distinct subpopulations of cells when ZEB1 was induced. At the end, the authors assess the involvement of ZEB1 in metastasis and a conclusion is made that ZEB1 is involved in bone metastasis. This is a potentially important topic and the work is of high quality. But the paper feels like lots of small individual parts that have been squeezed together in a paper and there are some fundamental issues with the existing manuscript.

We appreciate the reviewer's overall very positive statement and hope that the revision does a better job at connecting the "small individual parts".

- The paper seems to compare the ZEB1 ChIPSEQ profile from TNBC (ie MDAMB231 & Hs578Ts) with ER-alpha + models. I understand that this is because ZEB1 expression in parental MCF7 cells is probably too low. However, they use ZEB1 ChIPSEQ for all their comparisons (and snapshots) and they compare it with what happens to ER-alpha in MCF7 ZEB1 overexpressing cells. ZEB1 ChIPSEQ in MCF7 ZEB1 overexpressing cells is required otherwise a direct and fair comparison is not achieved. To this point, many of the Venn diagram overlaps are very small and might be what is expected by chance, probably because the authors are comparing two completely different cell contexts.

We have now done the requested ZEB1 ChIP-seq experiment with our own cells. All figures based on these results have been revised (Venn diagrams, GO analyses, motif analysis, and so on). There is now a considerable better overlap of chromatin binding sites, as illustrated with Venn diagrams.

- The authors conclude based on TF enrichment motifs that AP2G links ZEB1 overexpression to the novel ER-alpha binding sites (and less so, FOXA1). However, in fig 3h, they test the effect of AP2G knockdown at classic ER-alpha targets TFF1 and GREB1. Can the authors use a panel of 5-10 novel ER-alpha binding sites where AP2G motif is present (without FOXA1 motifs) and repeat the experiment to test the impact of AP2G specifically at these "novel" AP2G dependent, ZEB1- ER-alpha sites? Furthermore, the authors suggest that there are some novel ER-alpha binding sites where ZEB1 might bind directly (ie independently of AP2G) but this should be validated.

We have to agree and have now done the suggested experiments:

- Fig. 3i and Supplementary Fig. 5d-i show novel ERBSs at AP2 γ -dependent sites (with or without FOXA1 motif).
- Fig. 3j and the corresponding genome browser view in Supplementary Fig. 5j show AP2 γ -independent ER α binding and the effect of ZEB1 on that.

- Much of the paper relies of luciferase reporter assays which do not possess the biological relevance required.

While luciferase reporter assays may lack "biological relevance", our manuscript contains relatively few such experiments, which have value as orthogonal approaches to complement other types of data. Indeed, the vast majority of experiments and data are clearly not from reporter assays.

- The RNA-seq data suggests that a third of the genes in the genome are regulated by ZEB1. Surely this must mean that the threshold is not stringent enough?

Thank you for pointing that out. The original figure included all genes whose transcript levels changed with an FDR < 0.05. We now focus on fold changes of at least 1.3x (still with FDR < 0.05).

- The single cell work is elegant, but I am not sure what can be concluded from it or how it fits in with the rest of the paper. Where is AP2G expressed? Why are the genes identified from the 3D spheroid assays not explored in the different cell populations? At the moment, the single cell analysis only complicates the paper and dilutes the main conclusions.

- In the revised manuscript, we more explicitly explain why we did the scRNA-seq experiment and how this connects to the rest of the story. In contrast to RNA-seq, the scRNA-seq experiment is able to capture the heterogeneity of a cell population, even in a cell line. EpCAM being a relevant marker for hybrid EMT, we chose the time point accordingly. The experiment was done in medium containing a physiological dose of E2 and hence, when the functional interaction between ZEB1 and ER α is at work. We have now also tried to make it clearer why we followed up on certain genes that came out of the scRNA-seq experiment. For example, for CD151, we find that it is regulated by a complex interplay of ZEB1 and ER α , and that its knockdown dramatically impairs migration under conditions where ZEB1 and ER α can functionally interact (Fig. 6j,k). Similar functional results for other genes that had an interesting pattern in the scRNA-seq are reported in Supplementary Fig. 10.
- AP2 γ (and FOXA1) is expressed in both EpCAM^{high} and EpCAM^{low} populations (Supplementary Fig. 9h).

- "Genes identified from the 3D spheroid assays": These genes came out in a very different context (i.e. not by scRNA-seq). Identified because of their expression profile in the RNA-seq, their function was tested in invasion assays.

- The authors tentatively suggest that ZEB1 overexpression redirects metastasis formation to the bone (although bone morphogenesis is not an enriched pathway in figs 4 e+g or fig 7d). However, this is based on 2/5 vs 3/3 with no significant difference in metastatic index. Unfortunately with such small numbers, it is not possible to conclude anything with confidence.

These experiments indeed being at the heart of the story, we repeated them to strengthen the evidence. The experiments were set up with 5 additional animals each for the control cells and for cells with ZEB1 expression. Unfortunately, for several of them the experiment had to be terminated prematurely (too early to monitor the presence of metastasis) because ulcerations appeared. Nevertheless, this repeat did add several additional data points (xenografted animals), and the statistical analysis of the combined data now shows a robust difference: The metastatic index for bone is significantly higher for cells that express ZEB1 (Fig. 7a,b). We would also like to emphasize that it is already unusual for these types of cells to form bone metastases (they readily do it in the lung and other organs).

That "bone morphogenesis" or "osteoblast differentiation" are enriched GO terms was clearly highlighted (in red) in the original figures, and it still is in the revised ones.

Minor comments:

- The authors use ER-alpha CHIPSEQ for parent MCF7-V cells previously published (Ref #35) while for fig 2j-l, they use for comparison a newly generated ER-alpha CHIPSEQ with wild type MCF7-V cells? What is the difference between these two datasets and why not use the newly generated ER-alpha CHIPSEQ for both comparisons?

The difference between these two datasets is the method used for DNA fragmentation (published ChIP-seq with sonication and newer ChIP-seq with MNase digestion). The data are similar in the number of binding sites, but the newer ChIP-seq includes larger fragments (with potentially more preserved TF motifs). For the revisions we use the published ER α ChIP-seq data obtained with wild-type MCF7-V cells. The newer data (with MNase fragmentation) have been removed.

- RNASEQ and CHIPSEQ were carried out in biological duplicates. Since this is very few (for RNASEQ, potentially too few) replicates, it would be nice to see how the two replicates cluster together.

The heatmap of Fig. 5a shows it for the RNA-seq data, and the new Supplementary Fig. 2b shows it for the ChIP-seq replicates. These quality controls indicate that the respective duplicate samples are reasonably close to each other.

- Fig 1l (invasion kinetics quantification): difficult to distinguish between the panels. Also, did they all start from the same position at time 0? This is important to include this, otherwise small differences in starting size/area can be magnified over time, distorting the conclusion.

We transformed Fig. 1l into a bar graph, which makes it easier to show all underlying data points and error bars, and of course, we now include the 0 time point.

- Fig 2f (Motifs): are there any p-values associated with this?

Yes, there are. In the legend, we now explicitly refer to Supplementary Data 2, where p-values can be found.

- Fig 2m: Re-CHIP: would be good to show re-chip at some sites where ER-alpha binds alone or where ZEB1 binds alone? Or is GREB1 (-2kb) supposed to be one of those sites?

In fact, the reviewer rightly sensed that something was wrong there. We mistyped the GREB1 site. It was meant to be (and now is in the revised Figure) the -20 kb site where essentially only ER α binds. It gives the expected result, i.e. no difference between with and without ZEB1.

- Fig 5d: the authors state that “the error bars are not shown to simplify the graph”. Spheroid type work has a lot of variability – therefore, including the error bars is essential for interpreting this experiment.

We agree. The revised Figure 5d shows error bars. To make it more useful and less cluttered, we added the 0 time point but removed 48 and 96 h time points. The full dataset can be found in Supplementary Fig. 8d (with 0 time point and error bars).

Reviewer #2 (Remarks to the Author):

Ghahhari et al analyzed the coregulation of Zeb1 and ER α signaling in metastasis of breast cancer. The authors identified that during early EMT, when the cells are still in an epithelial/partial EMT state, Zeb1 enforces the expression of ER α target genes. Moreover, the two proteins interact and form a complex that involves AP2g and FOXO1 binding.

Acquisition of endocrine therapy resistance in ER+ breast cancer is a main obstacle leading to rapid progression and metastasis. Therefore a more detailed understanding is crucial and the presented analysis provide important mechanistic insight how tumor cells utilize EMT signals and ER activation for progression, providing sufficient interest for a larger community of Nature communications. The analyses are performed with high technical standards and skills. Although the first part of the manuscript is very straight forward, nicely written, providing crucial analyses to prove the hypothesis and derive solid conclusions, the second part does not follow a comprehensible logic of addressing the next key questions, is less convincing, lacks a thread and needs major revision. Therefore, the manuscript is very difficult to read and to understand. I have outlined my concerns in more detail below:

Figs 1-5 follow a specific logic that falls within a comprehensive scope. The authors identify that specifically ZEB1 in early pEMT supports ER α activation, whereas it downregulates ER α during later phases of EMT, regulate known and novel common target genes and form protein interaction in a ternary complex with AP2g and/or FOXO1. For the rest of the figures (6-7) the focus on the synergism between ZEB1 and ER α activation is lost. The rationale remains elusive why the authors do not separate cancer cells during ZEB1 induction into EPCAM hi/lo populations for scRNA-Seq. To me, only ZEB1-induced EMT is analyzed in this experiment, independent of ER α activation or expression. Moreover, it was not clearly specified, how identified differentially regulated CD151 (EPCAM hi vs. low) is now connected to the initial findings. In Figure 7 the authors then try to show that ZEB1/ER α activity is changing organ tropism during metastasis based on very unconvincing preliminary data in mouse models

(n=3). I suggest to leave the experiments from the last two figures out of the manuscript and focus more on the initial findings. A more thorough analysis of defining the interaction domains in ZEB1 and ER α would be beneficial.

We appreciate the generally very positive evaluation and the specific suggestions. Clarifying how the scRNA-seq experiments connect to the rest of story was clearly necessary. We performed additional experiments and revised the text to make this clearer. Please refer also to our response to a similar comment of reviewer #1.

We were already convinced when we submitted the first version of the manuscript that the xenograft experiments were an essential component since they tested predictions made from the various omics analyses. Since the peer review process pointed out important weaknesses in these data, we performed additional experiments. The data now support unambiguously our conclusion that ZEB1 favors the formation of bone metastasis (see our response to your point #9 below).

Our detailed response to the comment about the interaction domains is provided below in the context of point #4).

Specific points:

1. Fig. 1 b,c: how was this done exactly? Are these the cells from a, transfected with luci-constructs only? If not, it would be crucial to test this under DOX conditions for short and long treatment times separately.

The experiments of Fig. 1b,c were done by transient transfection (including ZEB1), as indicated in the legend. Results for stable MCF7-V-ZEB1 cells with and without DOX induction are now shown in Supplementary Fig. 1c,d.

2. Fig. 1h: Although the authors show that 293 cells show a moderate activation of the SBE reporter upon TGFbeta treatment, are 293 cells responsive to TGFbeta? Is this the right model to study this? Why not using the Dox-inducible system? 293 cells have a very high endogenous level of ZEB1 without stimulation

Well, our data show that HEK293T cells do respond to TGFβ. In contrast, they have negligible ZEB1 mRNA levels and no ZEB1 protein.

3. Fig. 2a and others: authors should provide a negative control showing the lack of enrichment of a non-related genomic region, like of a house keeping gene or x bp up-/downstream. What does “recruitment over input” mean? The ChIP data should be presented as “percentage of input”. It would be very interesting to understand whether also Zeb1 is recruited to these genomic regions in an ERa-activation-dependent manner. Is ZEB1 already present at the target regions in unstimulated, but “+DOX” situation and represses gene expression in the absence of ERa activation?

- Sorry we had not clearly stated that. This negative control is built into our ChIP assays and calculations. We have now added a statement to the paragraph Methods (values are relative to a non-binding region and input). In thinking about this again, we decided to change the Y-axis labels (for all graphs with ChIP results). What's important is what the values are relative to each other for a given graph. What % of input those are is actually irrelevant.
- Our focus was on how ZEB1 affects ERα, and not the other way around. It should be pointed out, though, that our ZEB1 ChIP assays were always done with cells grown in complete medium with physiological concentrations of E2 (now clearly stated both in Results and Methods).
- We explicitly point out in the revised manuscript that the RNA-seq revealed a smaller number of genes, which were up- or down-regulated by ZEB1 without active ERα (see also Fig. 4b). But again, the focus of our paper lies elsewhere.

4. Fig. S3: The interaction between ZEB1 and ER α is convincingly shown, so I wonder why the authors did not follow this path in a more in-depth analysis. I think it would be straightforward to test a construct that contains aa1-272 only (ZF1) in coIP in S3c. Moreover, subdividing ZEB1 into 3 non-overlapping fragments 1-272, 272-882 and 882-1061 would be helpful. In S3e: Why is ctrl+Zeb1 not activating the reporter if E2 was added, like in S3b? S3e: WHY was full-length ER α not included in the assay? Why is HBD alone sufficient to activate ERE? Is it binding to the ERE elements in absence of the DNA-binding domain? I don't see why the authors claim that the F-domain should be involved in ZEB1-ER α binding? These experiments need to be more concise or left out because they are inconclusive. Moreover, 293 cells express high levels of Zeb1, so the authors may consider a knockdown of Zeb1 when introducing the ER-domain constructs (S3e). Fig. S3f: input controls of GST/His-tag

proteins are missing. Fig. S3a: It needs to be indicated whether the construct "ZF1-ZF2" contains point mutations of the HD or a deletion. In case of the latter, a deletion should be indicated by interrupted/joined parts of the N and C-terminal fragments. Is ZF1-ZF2 the same as dHD? The nomenclature in a and b should be similar! Fig. S3c: I guess the IP was done with anti-HA, so it should be indicated in the figure.

- It seems that the reviewer misinterpreted our results, also because we overlooked an important typo in the legend for panel e. It should have been "Gal4 luciferase reporter", not "ERE luciferase reporter". We would like to refer the reviewer back to the Supplementary Results (not just the Supplementary Figure 3). Hopefully, things will be clearer now.
- Our mapping was based on loss-of-function (interaction and ER α stimulation) experiments with ZEB1 truncations and Gal4-ER α fusions (transcriptional regulation). The former indicated that ZF1 may be involved and the latter showed that the only chimeric protein to be stimulated was the HBD including the F domain. This is why we directly tested whether ZF1 interacts with the ER α F-domain. All of this was explicitly stated in Supplementary Results.
- These results clearly show that things are more complicated (which is what we stated in the Results section). Digging deeper would not necessarily provide more useful insights.
- Suppl. Fig. 3f does not contain input controls because these were carefully dosed recombinant purified proteins. The used protein quantities were added to the legend.
- We revised the scheme of panel a to indicate that the homeodomain mutant is a deletion mutant and we unified the nomenclature across the different panels.
- Panel c: no, the IP was actually done with an anti-ZEB1 antiserum, which apparently recognizes all these truncation mutants. This was originally already indicated in the Methods, but to make it more obvious, it is now also mentioned in the legend of this Figure.

5. Fig. 5d: The authors claim that invasion of ZEB1 expressing cells is inhibited when MUC16, DSCAM etc. are knocked down. This is very unconvincing as only marginal differences are observed and indicating that the cooperation of the two proteins is maybe not acting mainly on invasion. Maybe the data need to be shown in more simplified manner with less parameters

Indeed, because the figure was too difficult to read, we revised it substantially. Please refer to our response to the last comment of reviewer #1.

6. Figure 6/S8: Like I indicated above, I am confused by the twist in the analysis by now dissecting EPCAM hi/lo cells. At which time-point was this done, early vs late DOX treatment and EMT? How does this fit to the initial finding that ER is downregulated in long term Zeb1 expressing cells. How does this relate to ER α ? Here only different transition states of EMT induction are analyzed

independent of ERα activation. Is there E2 treatment involved? How do the authors come up with focusing on integrins?

Clearly, this twist was not well explained. We have substantially revised the manuscript in this regard (also with additional experiments). Please refer to our response to the comment of reviewer #1 starting with "The single cell work is elegant, but".

7. Fig 6i-m/S8g,h: CD151 appears like a randomly picked target gene and the connection to ZEB1/ERα is enigmatic. Moreover, why is now sensitivity to EGFR inhibition analyzed? The effects in Fig. 6 k,l show that they are likely the result of two independent additive effects, but where is the connection/synergism?

In the revised manuscript, we tried to clarify why we did scRNA-seq experiments and why we picked CD151 (and several others) for follow-up experiments. We removed the EGFR inhibitor experiments from the revised manuscript.

8. Fig. 6m: The Kaplan-Meier plots need to be evaluated in combination with single effects of Zeb1 hi/lo (Fig. S1) and CD151 hi/lo (not yet shown): is there an improved separation of the two groups/survival benefit? It looks like that mainly Zeb1 high/low is a good prognostic marker independent of CD151.

We have now added a KM plot for CD151 by itself in Supplementary Fig. S10i.

9. Fig 7: Since the cells are analyzed in vivo for 10 weeks under constant DOX supply, it is difficult to assess whether the effects are due to an early pEMT, when ZEB1 cooperates with ERα or due to a late EMT phase, when ERα is already downregulated. The analysis of tumor size by BLI is not appropriate, especially since the images show that they are of similar size. Since the ZsGreen and ZEB1 cells have been infected separately, the luciferase expression might be at different level. Why was only E2 treatment but no untreated control included? Fig S7f: The conclusion of changes in organ tropism are very unconvincing with the assays presented. A colonization experiment would help to define whether cells injected into the circulation will preferentially colonize the bone or would also create more lung mets when ZEB1 is induced. Again: what is the connection to ERα? In this experiment even no E2 was added, so no ERα activation is involved?

- Complete EMT is extremely unlikely, and cancer cells have been proven to go through partial EMT states instead. With our new analysis of primary tumors and bone metastatic lesions (using IF; see Fig. 7c-h and Fig. S11b-f), we observe that ERα and EpCAM expression are maintained in the presence of ZEB1 in the majority of cells. We discovered clusters of cells that showed both EpCAM and vimentin, indicating incomplete EMT states. This means that, while ZEB1 can produce a fully mesenchymal state *in vitro*, ZEB1 expression in physiological conditions only promotes a partial EMT, and ERα is still present in these cells.
- We now provide more details (in Results, Methods, and Figure legends) on how we measured tumor sizes and calculated metastatic indices.
- ZsGreen and ZEB1 cells were infected separately using the same pool of cells that had first been infected with a luciferase reporter. Therefore, luciferase expression levels are unlikely to be different between ZsGreen and ZEB1 cells.
- As mentioned in our response to reviewer #1, we have done additional xenograft experiments and now provide considerably stronger and statistically significant data to claim that ZEB1 expression reprograms cells towards bone metastasis (with similar degrees of metastasis in other organs such as lung).

- MCF7 cells are E2-dependent, and tumor development cannot occur when E2 is not present. As a result, E2 is administered to mice in the form of E2 pellets surgically implanted in their backs.

Minor points:

1. "DMFS" is misspelled in the text (DMSF). corrected
2. All Western blots lack Mw standard indicators added
3. Fig. S8h: there is a labeling error, as Dox/sh treatments are similar between left and right. Maybe the first 2 lanes are -DOX and the last 2 are +DOX? corrected

Reviewer #3 (Remarks to the Author):

Review NCOMMS-21-17328

In This manuscript, Ghahhari and colleagues investigate the transcriptional cross talk of ZEB1 with Estrogen receptor Alpha (ER α) in the context of hybrid/transitory EMT. The manuscript begins with a series of biochemical approaches to characterize ZEB1 and ER α at the level of transcriptional regulation (figure 1), chromatin regulation (Figure 2) and protein interaction (Figure 3). They then expand this observation using RNAseq (Fig 4-5) and single cell RNAseq (fig 6). Finally, they perform additional in vivo or ex vivo studies to link these observations to phenotypical changes occurring in metastasizing cancer cells. Overall, these observations are extremely interesting and reveal a previously unknown axis of regulation for ER α . The novelty and limitation of this study stems from the transitory nature of these hybrid phenotypes and their potential role in breast cancer biology. I believe this is potentially an interesting study but I also think that the manuscript requires some work before it could be considered for publication. Some of the analysis are too preliminary and the use and integration of additional datasets from alternative models is not always well justified.

We appreciate the reviewer's overall very positive evaluation and hope that we have addressed the paper's shortcomings in our extensive revision.

Major Concerns:

1- The cross talk between ER and ZEB1 appears to have an "expiration date", which is represented by the mature EMT transition. The author uses a ZEB1 inducible model. What happens if the DOX is turned off at various stages if ZEB1 induction? When EMT becomes irreversible?

These are all interesting questions, but they go beyond the scope of this study. Our aim was to investigate the impact of ZEB1 on ER \$\alpha\$ at earlier stages of EMT.

2- Following, this, it is essential to understand when and why ZEB1 would be activated in luminal BRCA. The model (figure 8) does not offer any indication, and albeit the authors might think this is not the point of the study, most of the clinical data presented by the same authors suggest that high ZEB1 is good because it reinforce the ER α signalling network. Either the authors focus the paper on the mechanistic aspect, or it should justify this apparent contrast (as ZEB-ER should be short lived and linked to earlier and more productive microdissemination

This again requires more dynamic experiments in vitro and in vivo comparing the different stages of ZEB1-induced EMT on ER \$\alpha\$ signaling. It would be interesting to use patient-derived xenograft experiments and test the discovered mechanism in tumors and metastatic lesions of patients with

tumors at different stages of breast cancer. All highly interesting but well beyond the scope of this study.

3- The effect of ZEB induction on “invasion” in the 3D models (Fig. 1K,L) should account for potential difference in basal cell proliferation induced by ZEB1 activation (which emerge from figure 1C). Some proliferation 2D/3D measurement would give this reviewer more confidence this is truly invasion and not increase proliferative pressure.

All migration and invasion assays were performed in medium with low serum and mitomycin was included to inhibit cell proliferation (as mentioned in the revised Methods). The new proliferation assay in Fig. 6j indicates that ZEB1 does not impact proliferation of cells.

4- Figure Supp 1N. Overstatement, these data alone do not support the statement “ZEB1 sensitize...”, as ZEB1 activation increases the G0/G1 fraction

We adjusted the statement regarding what's now Supplementary Figure 1p in the Results to avoid overstating.

5- Can the author add more details in the text around figure 2? ER CHIP seq was done in MCF7-V-ZEB1. Is this short term, long term induction? Or something else?

As indicated in the Methods, it is short term. With long-term ZEB1 expression, there wouldn't be any ER α left.

6- Overlapping ER α from MCF7 ZEB1 (I assume) with ZEB1 from MDAMB231 is not appropriate. This reviewer believes that a ZEB1 ChIP-seq dataset from the same model is required prior to publication (and should be subbed in the analysis to the current one).

Yes, very well taken comment. As also indicated in our response to reviewer #1, we have now done the requested ZEB1 ChIP-seq experiment with our own cells. All figures based on these results have been revised (Venn diagrams, GO analyses, motif analysis, and so on).

7- Page 11 “ We decided.....(Fig 3B). It is not clear from what models the AP2 and FOXA1 dataset come from.

We revised the text to indicate the sources of these datasets.

8- The FOXA1 KD experiment is very tricky as most cells would die in the process. Indeed, only 50% mRNA suppression was achieved. Hard to interpret.

As can be seen in Supplementary Fig. 5a, the knockdown of FOXA1 is only partial, which might explain why cells survive the selection (for the marker carried by the lentivirus for expression of the specific FOXAA1 shRNA). Thus, the functional impact of this knockdown can very well be considered.

9- Page 12: when the authors refer to “protein complexes” it would be more appropriate to disclose how these experiments were done and then rewrite these sections. Are these interactions occurring in the cytoplasm, nucleus or on the chromatin? Most of FOXA1 is bound to chromatin, AP2 is also considered by some a Pioneer factor, mostly chromatin bound. ER is much more dynamic. What is the authors interpretation of their data? Are these complexes formed in solution or are these data reflecting a more “additive” scenario with FOXA1, AP2 already bound and ER-ZEB1 coming later on in response to ligands or other stimuli?

Hopefully the reviewer will find that the revised text (Results and Discussion) makes it clearer. Having said that, further investigations would be required to dissect the exact order of events.

10- Fig 4. More details on how long DOX was given is required for a correct interpretation of the data.

Please refer to our answer to your point #5.

11- Page 15. "this suggest the existence....EMT stages". How robust are these data considering the drop-out effect of scRNAseq? Can the authors show saturation analysis? With scRNA data, often is a question of sensitivity.

The saturation plots are now included in Supplementary Fig. 9c,d.

12- The EGFR inhibitor experiments are vastly exploratory. If EGFR dependency is as short lived as the ZEB1-ER cross talk, when would this vulnerability become valuable (if detectable at all) in a clinical scenario?

We removed the EGFR inhibitor experiments from the revised manuscript.

13- The in vivo experiments would require some pathology characterization. For how long was DOX induced here? Why is it that controls n=4 and ZEB n=3? We noted that one of the control in Fig 7B has a striking load to the bone (indeed the pVal is not significant). The discussion of figure 7 should be tone down significantly as the experimental designed is not really conclusive (fig. 7E quantification?)

We have done additional xenograft experiments (Fig. 7 and Supplementary Fig. 11) and added an analysis of primary tumors and bone metastatic lesions (using IF; see Fig. 7c-g and Fig. S11b-f). Please see also our response to point #9 of reviewer #2.

Indeed, one of the control animals was a serious outlier, but with additional data points, we can now unambiguously conclude that ZEB1 promotes the formation of bone metastases.

Regarding Fig. 7e (now revised Fig. 7i), these images should be considered a qualitative indication that ZEB1-expressing cells show bone tropism. These data obtained with an orthogonal approach are complementary to the xenograft results and the transwell assays (other panels of Fig. 7).

Overall, I'd be keen to see a vastly reformatted and streamlined manuscript which focus on the main finding without unnecessary deviations or forced clinical relevance.

Minor points

Fig. 1B. The veh results are normalized to 1. It is therefore impossible to judge the effect of ZEB1 induction on basal levels of ERE driven transcription (which is apparent in Fig 1C).

We revised the figure (the bars now show RLU, i.e. relative to the internal transfection control Renilla without normalization to the vehicle control).

ChIP results from Figure 2D suggest ZEB1 induction as a quantitative effect on ER binding (not an ON/OFF). Suggesting that even in the absence of ZEB1, these loci recruit, in some cell, ER.

Yes, that's correct for most sites. However, we had also presented ChIP data for sites, which are unlocked ("ON/OFF") by ZEB1. These results were/are presented in Fig. 2h.

Reviewer #4: (Remarks to the Author):

In this manuscript, Nastaran Mohammadi Ghahhari et al. analyzed EMT-related interaction of ER α , ZEB1 and other regulatory factors. They identified and characterized the functional interaction of ZEB1 and ER α by using MCF7 and some cell line systems. They also conducted ChIP-seq analysis and invasion assays for confirming transcriptional programs and EMT phenotypes. They found that ZEB1 modulate ER α signaling by recruiting novel ERBSs and sharing the regulatory elements. The authors additionally performed scRNA-seq and identified CD151 as therapeutic targets during early/hybrid EMT stage. They finally conducted xenograft/ex vivo assays to unveil relationship between the ZEB1-associated program and bone metastasis. The authors showed a lot of results using MCF7-based systems. However, they should clearly describe whether the obtained results could be represented in real cancer tissues. There are some points that need to be addressed to improve the manuscript as below;

Points;

1. The authors need to explain the detailed experimental condition, strategy and purpose of scRNA-seq. In scRNA-seq analysis, they should indicate which cells come from EPCAM high or EPCAM low fractions of FACS sorting and show expression levels of ZEB1 in each cell. I'm not sure which time course samples were used for scRNA-seq experiments. I also think that bulk RNA-seq analysis is enough for the comparison of the two populations according to EPCAM expression.

We have substantially revised the manuscript in this regard (also with additional experiments) to make it clearer why we did these experiments and what can be learned from them. Please refer also to our response to the comment of reviewer #1 starting with "The single cell work is elegant, but".

2. Are cells at the hybrid EMT state in real clinical samples of breast cancers? The authors should add information of the obtained results in clinical samples.

To support the idea that hybrid EMT states exist, we now provide more background information, notably in the Introduction. To investigate this experimentally is beyond the scope of our current study.

Minor points;

1. In Fig. 6a, it is not easy to associate colors and cluster numbers. To easily understand Figs. 6c and e, the authors should add the cluster numbers to the UMAP or some description to the figures.

We have modified the Figure accordingly.

2. They defined MKI67 as epithelial cell markers in scRNA-seq analysis. To my knowledge, MKI67 is a proliferation marker and their expression patterns are dependent on cell cycle state. The authors should describe the association between the findings of scRNA-seq and cell cycle state of each cluster if they did not perform regression of cell cycle effects.

Thanks for pointing that out. We have removed (M)KI67 from the revised manuscript.

Regarding the cell cycle state of each cluster, this is less of an issue with cell lines than with primary cells and tissues. Indeed, cell cycle genes (rather: their transcripts) were not excluded from our analysis pipeline and yet are not associated with specific clusters.

3. In p. 13, please specify the number "> 1,000".

By going back to the data and the relevant Figure (Fig. 4b), it turns out it is actually "hundreds", not more than one thousand. The exact numbers are not relevant can of course be found in the Venn diagram.

4. In Fig. 8, the authors represented the overview of the obtained results. I recommend that they add information of EMT stage (early, hybrid and late) and other key molecules which were identified in this study, such as CD151 if possible.

The summary scheme of Fig. 8 has been revised, although we prefer to keep it relatively "light".

REVIEWERS' COMMENTS

Reviewer #1 (Remarks to the Author):

The authors have taken a comprehensive and careful approach to address my concerns. The inclusion of the new datasets and new analyses have addressed all of my original concerns and I am happy with the revised version of this manuscript.

Reviewer #2 (Remarks to the Author):

The authors appropriately addressed most of my concerns. However, for two small points they could not sufficiently improve the manuscript and I suggest to fix these before final acceptance of the manuscript. Otherwise, I recommend publication of this nice piece of work.

1. Suppl. Fig. 3f: The authors state in the rebuttal that “Suppl. Fig. 3f does not contain input controls because these were carefully dosed recombinant purified proteins”. Such GST pulldown experiments should always show an input control, especially since no signal was observed in any lane of the His blot. I wonder whether the protein is present at all.

When checking the experimental details, it is becoming even more puzzling. As far as I understood, the GST pull down was done as follows: GST-F domain from ERa and a GST-Ctl non-related fusion protein was bound to beads separately. To both tubes His-tagged ZF1 (ZEB1) was added. After pull-down and gel separation, the both GST proteins should be visible in the anti-GST blot. They are, but to very different amounts, meaning that they are NOT properly titrated. In the anti-His blot, I would expect a band corresponding to ZF1 when binding to the F-domain occurs. However, in the corresponding lane 3, there is no His protein detected, neither in the lane of the negative control. So, what does this experiment tell us? The authors claim that they could not confirm direct binding. This can have many reasons, but using only these two fragments in the GST pull down is not sufficient, especially since they were not tested in the IP experiments simultaneously (maybe ZEB1 and ERa do not bind directly). Again, the figure S3 only shows that for interaction ZEB1 ZF1 is crucial, but whether it is sufficient or not is not solved.

For Fig. 3f, a second gel with 1:100 (or so) of input should be shown to observe that the proteins were properly expressed but were not able to bind. If no additional experiments are added, I would remove the pulldown in Suppl. Fig.3f since it is inconclusive.

2. MW standards for all Western blots: With my comment, I was not aiming for adding the precise Mw of the detected proteins to the Blots, but to include the position(s) of the bands of the Mw standard (Marker) that was run in a separate lane. Alternatively, this can be added to the uncropped versions of the blot.

The reason for this request becomes obvious when comparing the blots from Suppl. Fig. 3c. It is puzzling that all ZEB1 deletion constructs should have the same Mw as the full length protein (220 kDa), when 100-200 aa are missing. This needs clarification.

Reviewer #3 (Remarks to the Author):

The authors have done a great job in adding experiments and analysis to the manuscript. I still believe there is a missed opportunity as the temporal dynamic of the ZEB-ER cross-talk is a central point.

Reviewer #4 (Remarks to the Author):

In the revision, the authors have improved the manuscript and addressed the previous concerns including those about scRNA-seq analysis. The reviewer still thinks that they should at least add the discussion whether the identified molecular programs really exist in individual cancer cells of breast cancer tissues. They mentioned that "Recent studies have reported the coexistence of epithelial and mesenchymal-like cell states, as well as differential hormone receptor expression, within the same tumor cells, demonstrating the occurrence of hybrid EMT states in cancer patients" (line 633) so that they can compare the results (molecules) of this study with those from the previous reports.

Point-by-point response to reviewers' comments

Note that our responses to the reviewers' comments are in blue.

REVIEWERS' COMMENTS

Reviewer #1 (Remarks to the Author):

The authors have taken a comprehensive and careful approach to address my concerns. The inclusion of the new datasets and new analyses have addressed all of my original concerns and I am happy with the revised version of this manuscript.

Thank you, much appreciated.

Reviewer #2 (Remarks to the Author):

The authors appropriately addressed most of my concerns. However, for two small points they could not sufficiently improve the manuscript and I suggest to fix these before final acceptance of the manuscript. Otherwise, I recommend publication of this nice piece of work.

1. Suppl. Fig. 3f: The authors state in the rebuttal that "Suppl. Fig. 3f does not contain input controls because these were carefully dosed recombinant purified proteins". Such GST pulldown experiments should always show an input control, especially since no signal was observed in any lane of the His blot. I wonder whether the protein is present at all.

When checking the experimental details, it is becoming even more puzzling. As far as I understood, the GST pull down was done as follows: GST-F domain from ERa and a GST-Ctl non-related fusion protein was bound to beads separately. To both tubes His-tagged ZF1 (ZEB1) was added. After pull-down and gel separation, the both GST proteins should be visible in the anti-GST blot. They are, but to very different amounts, meaning that they are NOT properly titrated. In the anti-His blot, I would expect a band corresponding to ZF1 when binding to the F-domain occurs. However, in the corresponding lane 3, there is no His protein detected, neither in the lane of the negative control. So, what does this experiment tell us? The authors claim that they could not confirm direct binding. This can have many reasons, but using only these two fragments in the GST pull down is not sufficient, especially since they were not tested in the IP experiments simultaneously (maybe ZEB1 and ERa do not bind directly). Again, the figure S3 only shows that for interaction ZEB1 ZF1 is crucial, but whether it is sufficient or not is not solved.

For Fig. 3f, a second gel with 1:100 (or so) of input should be shown to observe that the proteins were properly expressed but were not able to bind. If no additional experiments are added, I would remove the pulldown in Suppl. Fig. 3f since it is inconclusive.

2. MW standards for all Western blots: With my comment, I was not aiming for adding the precise Mw of the detected proteins to the Blots, but to include the position(s) of the bands of the Mw standard (Marker) that was run in a separate lane. Alternatively, this can be added to the uncropped versions of the blot.

The reason for this request becomes obvious when comparing the blots from Suppl. Fig. 3c. It is puzzling that all ZEB1 deletion constructs should have the same Mw as the full length protein (220 kDa), when 100-200 aa are missing. This needs clarification.

1. We have followed the reviewer's advice and removed the GST pull-down. This is indeed justifiable also by the fact that the evidence here is only negative (i.e. absence of interaction). We have adapted the main text and the Supplementary Information accordingly.

2. The original uncropped blots for Supplementary Fig. 3c now show part of the MW marker, in addition to the predicted MW of the ZEB1 truncation mutants (see Source Data file). The latter are almost all within 10-20 kD of the wt protein, which is a large protein (220 kD). This is unfortunately not in a size range where separation is particularly good. We have added a statement to the legend of this figure to indicate that MW markers are available in the Source Data file.

Reviewer #3 (Remarks to the Author):

The authors have done a great job in adding experiments and analysis to the manuscript. I still believe there is a missed opportunity as the temporal dynamic of the ZEB-ER cross-talk is a central point.

We appreciate the positive feedback and understand that the temporal dynamics would be great to see. Unfortunately, these are challenging experiments, which we have to leave to future investigations.

Reviewer #4 (Remarks to the Author):

In the revision, the authors have improved the manuscript and addressed the previous concerns including those about scRNA-seq analysis. The reviewer still thinks that they should at least add the discussion whether the identified molecular programs really exist in individual cancer cells of breast cancer tissues. They mentioned that "Recent studies have reported the coexistence of epithelial and mesenchymal-like cell states, as well as differential hormone receptor expression, within the same tumor cells, demonstrating the occurrence of hybrid EMT states in cancer patients" (line 633) so that they can compare the results (molecules) of this study with those from the previous reports.

Yes, the coexpression, which we demonstrate and discuss for individual cells in our histological analysis, can also be seen in the scRNA-seq analysis. To state that more explicitly, we have added some comments to the relevant part of the Results.